# GEOMETRY MEETS VISION:
## REVISITING PRETRAINED SEMANTICS IN DISTILLED FIELDS

## ABSTRACT

Pretrained semantics from large vision models have enabled major advances in open-vocabulary robot policies, e.g., in manipulation and navigation. However, a striking lack of consensus on the performance and effects of fine-tuning these vision encoders remains a significant challenge. For example, some papers claim that (task-specific) pretrained encoders outperform general-purpose semantic encoders (e.g., DINO) or that fine-tuning vision encoders improves performance, while others claim the exact opposite. In this work, we seek to address these long-standing divisions through a principled examination of pretrained semantics from vision encoders in robotics. We hypothesize that the inconsistencies in prior work arise from a fundamental lack of insight into the feature content of these vision encoders. Hence, we undertake a systematic study of pretrained semantics in distilled fields to uncover their salient components with the goal of identifying a framework that explains previously contradictory claims. Specifically, we ask: *what do the semantic features of robotics vision encoders contain?*—and consider visual-semantic encoders (like DINO) and geometry-grounded encoders (like MUSt3R/VGGT). Notably, we find that these encoders attend to different features in their image inputs. While visual-semantic encoders prioritize object/part-level semantic decomposition, geometry-grounded encoders may discard this information to focus on more structural components, such as edges and corners. This observation can be described by catastrophic forgetting of core semantic information, which worsens with increased fine-tuning. We validate these findings in two major robotics problems: semantic object localization and radiance field inversion, using distilled fields as a testbed. We observe results consistent with the internal contents of the semantic features of these encoders, highlighting the strong explainability afforded by internal probes. For semantics-focused radiance field inversion, we propose a novel framework SPINE using distilled semantics for coarse inversion followed by a fine inversion procedure with photometric-based optimization, *without* an initial guess, demonstrating its superior performance compared to competitive alternatives. Further, our results suggest that geometry-grounding could offer potential benefits if catastrophic forgetting is controlled.

## 1 INTRODUCTION

Large foundation models have driven rapid advances in open-vocabulary robot policies, enabling robots to perform complex, multi-stage tasks, entirely from natural-language instructions; see (Firoozi et al., 2025) for a detailed review. Essentially, all state-of-the-art foundation models rely on pretrained vision encoders for robust processing of images into intermediate conditioning inputs for these foundation models. Despite the importance of pretrained vision encoders, a striking lack of consensus exists on the properties of pretrained semantics from these encoders, ranging from their capabilities and performance to the effects of fine-tuning. While some prior works (Chi et al., 2025) claim that (task-specific) pretrained encoders outperform general-purpose semantic encoders (CLIP/DINO) or that fine-tuning vision encoders hurts performance (Karamcheti et al., 2024; Huang et al., 2025), other work (Kim et al., 2025) claims the exact opposite. These contradictory findings pose a major challenge to researchers and practitioners alike. We hypothesize that the secret to understanding these seemingly inconsistent claims lies in the internal content of the semantic features of these encoders. Notably, in robotics, there is a prominent lack of research examining the relative

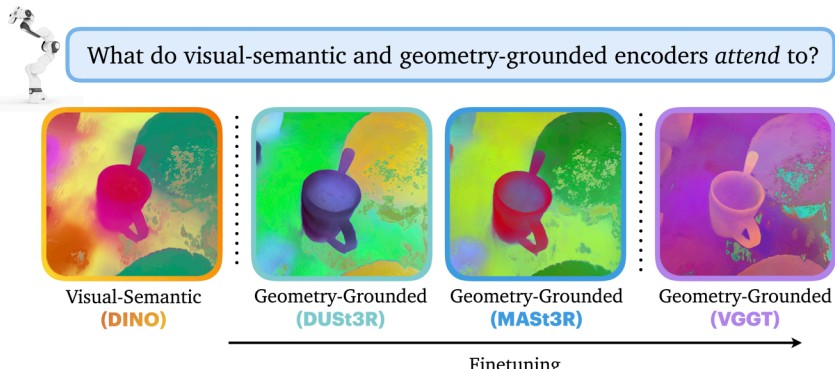

Figure 1: We revisit pretrained semantics in distilled radiance fields, probing the semantic features of robotics vision encoders to uncover their internal content. We identify what these vision encoders attend to and draw connections between finetuning and generalization of these encoders, particularly for visual-semantic and geometry-grounded vision encoders.

composition of semantic features across vision encoders to inform better integration of these encoders into robotics pipelines, with a few exceptions from the computer vision community, such as DINO. In this work, we take the first steps towards shedding more light on the content of these features for better explainability, which is essential to future research progress.

We undertake a systematic study of pretrained semantics to uncover the salient components of semantic features of robotics vision encoders. In this study, we consider the most widely-used vision encoders, grouped into two categories: *visual-semantic* encoders—e.g., DINOv2/v3 (Oquab et al., 2023; Siméoni et al., 2025)— and *geometry-grounded* vision encoders—e.g., DUSt3R (Wang et al., 2024), MASt3R (Leroy et al., 2024), MUSt3R (Cabon et al., 2025) and VGGT (Wang et al., 2025)). We define *geometry-grounded* encoders as vision models that have been fine-tuned from a base visual-semantics encoder on a task that provides geometric supervision, such as 3D reconstruction. We choose to explore these encoders based on the important role played by geometry and semantics in robotics tasks. Specifically, we ask: *what do the semantic features of these vision encoders contain?* We leverage distilled radiance fields as a testbed for this study to control the effects of confounding variables. Then, we analyze the feature content to draw valuable insights into the performance of these features in downstream applications. We find that these vision encoders attend to different characteristics in their input images. While visual-semantic encoders focus on preserving object/part-level semantic information that distinguishes between different classes of objects, geometry-grounded encoders may discard this semantic information in favor of structural information that emphasize fine-grained edges, corners, and other spatial details.

These findings hold significant explanatory power. First, our findings suggest that general-purpose semantic encoders (such as DINO) would outperform task-specific encoders (such as DUSt3R, MASt3R, etc.) in generalist robot manipulation across a broad range of tasks, as observed in prior work (Kim et al., 2025; Huang et al., 2025). On the other hand, task-specific encoders would likely outperform general-purpose semantic encoders on dexterous robot manipulation tasks, given sufficient training data coverage, as observed in existing work (Chi et al., 2025). Second, our findings highlight that finetuning might degrade generalization by replacing more generalizable feature content with task-specific information, which resolves the discrepancies in the results of prior work, e.g., (Karamcheti et al., 2024; Kim et al., 2025; Huang et al., 2025).

We validate these findings in two key robotics problems in radiance fields: semantic localization and radiance field inversion. To enable semantics-oriented radiance field inversion, we introduce SPINE for initialization-free inversion, a critical challenge in prior methods (Yen-Chen et al., 2021; Chen et al., 2025). SPINE directly leverages embedded pretrained semantics to compute: (i) coarse pose estimates using a co-trained semantic field which maps semantic features to a distribution over camera poses and (ii) fine pose estimates by refining the coarse solution through novel-view synthesis in radiance fields and robust perspective-$n$-point optimization. Across all our experiments, we observe results consistent with our earlier findings. Concretely, we observe a trend where more finetuning leads to further catastrophic forgetting, hurting performance especially in radiance field inversion, a

task that requires strong global semantic understanding, while performance on semantic localization remains similar.

## 2 RELATED WORK

**Pre-trained vision encoders.** Foundation models have shown impressive capabilities as vision backbones, enabling zero-shot deployment in many downstream tasks without requiring specific finetuning. Increasingly, robot policies have embedded semantics from CLIP and DINO into radiance fields to enable language-conditioned robot manipulation (Rashid et al., 2023; Shen et al., 2023; Shorinwa et al., 2024b), mapping (Shorinwa et al., 2025), and object localization (Yin et al., 2025). One approach to building general-purpose pretrained foundation models is through self-supervised learning. For example, DINOv2 introduces self-distillation to learn task-agnostic image- and pixel-level features, which DINOv3 subsequently scales to larger models and datasets (Oquab et al., 2023; Siméoni et al., 2025). VGGT (Wang et al., 2025) extends this paradigm to 3D reconstruction tasks, finetuning DINO on a suite of 3D tasks to ground DINO features with geometric supervision. On the other hand, DUSt3R and its descendants (Wang et al., 2024; Cabon et al., 2025) finetune vision transformers to solve 3D vision tasks, e.g., multi-view stereo reconstruction. MUSt3R (Cabon et al., 2025) extends the prediction from pairs to multiple views, and MASt3R Leroy et al. (2024) extends it to image matching. While all of these models serve as useful feature extractors for 2D and 3D vision tasks, prior work has observed a significant degree of variation in their performance. In this work, we probe the features of pretrained vision encoders to shed some light on their observed performance and motivate future research based on insights on their feature content.

**Distilled Semantics in Radiance Fields.** Radiance fields marked a notable breakthrough in 3D scene reconstruction, achieving photorealistic image rendering and novel-view synthesis entirely from RGB images. NeRFs and Gaussian Splatting (GS) (Kerbl et al., 2023) have been widely applied in robotics, e.g., robot planning (Adamkiewicz et al., 2022; Chen et al., 2024), localization (Yen-Chen et al., 2021; Maggio et al., 2022), and manipulation (Kerr et al., 2022; Weng et al., 2022; Lu et al., 2024; Chen et al., 2025; Michaux et al., 2025). In this work, we explore pre-trained vision encoders in distilled radiance fields, analyzing their feature contents to draw generalizable conclusions for robotics applications.

## 3 VISUAL-GEOMETRY SEMANTICS IN DISTILLED RADIANCE FIELDS

To uncover the semantic feature contents of the robotics vision encoders, we distill their features into radiance fields, providing a controlled environment for rigorous evaluation. We consider the following foundation vision models: DINOv2, DINOv3, VGGT, DUSt3R, MASt3R, and MUSt3R. Although we distill CLIP features into the radiance field, we do not directly compare against CLIP features. Rather, in line with prior work Kerr et al. (2023), we augment each of the semantic encoders with CLIP features to enable open-vocabulary interaction. To jointly learn the CLIP embeddings alongside the other semantic features, we use a shared hashgrid to allow the flow of gradients between both semantic features.

**Extracting Pretrained Visual-Geometry Semantic Features.** We extract ground-truth pretrained semantic embeddings for each image from each of the vision encoders. Given a query image $\mathcal{I} \in \mathbb{R}^{H \times W \times C}$, each of the encoder outputs a semantic embedding $f \in \mathbb{R}^{H \times W \times d_s}$, where $d_s$ is the dimension of the feature space and is dependent on the specific encoder. For computational efficiency, we preprocess the entire dataset prior to training. We follow standard procedure in extracting these features, feeding the input images into the encoders and caching the resulting embeddings.

**Distilling Semantics into Radiance Fields.** We learn a semantic field $f_s : \mathbb{R}^3 \mapsto \mathbb{R}^{d_s}$, which maps a 3D point $\mathbf{x}$ to features extracted from the vision encoder, $f_s(\mathbf{x})$. Alongside, we learn a semantic field $f_l : \mathbb{R}^3 \mapsto \mathbb{R}^{d_l}$ that maps 3D points to the shared image-language embedding space of CLIP, where $d_s$ and $d_l$ denote the dimensions of the embedding space of the vision encoder and CLIP, respectively. We train a semantic field for CLIP features to enable downstream open-vocabulary tasks. For effective co-supervision of both semantic fields, the encoder and CLIP semantic fields share the same hashgrid encodings (i.e., base semantics),

associating their semantic embeddings with the same visual and geometric features, illustrated in Figure 2. We reiterate that this technique is standard in distilled fields.

Given a dataset $\mathcal{D}$ of images and associated camera poses, we render images in the semantic space by back-projecting RGB images using the estimated depth from the radiance field to reconstruct a point cloud in the local camera frame. We subsequently compute the vision encoder and CLIP semantic features for the constituent points with $f_s$ and $f_l$, respectively.

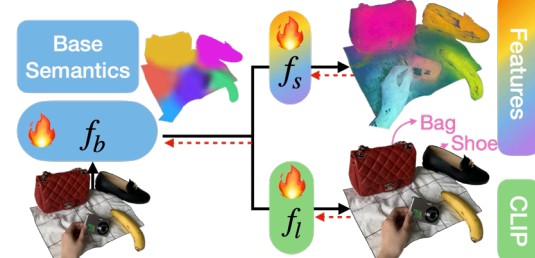

Figure 2: **Semantics distillation architecture**, showing co-supervision of CLIP with the other semantic encoders via the base semantics module.

During training, we optimize the parameters of $f_s$ and $f_l$ simultaneously with the visual attributes of the radiance field using the loss function: $\mathcal{L} = \mathcal{L}_r + \sum_{\mathcal{I} \in \mathcal{D}, c \in \{s,l\}} \|I_{f,c} - \hat{I}_{f,c}\|_F^2 - \sum_{\mathcal{I} \in \mathcal{D}, c \in \{s,l\}} \mathrm{csim}(I_{f,c} - \hat{I}_{f,c})$, where $\mathcal{L}_r$ denotes the RGB loss components of the base radiance field, $I_{f,c}$ and $\hat{I}_{f,c}$ denote the ground-truth and rendered semantic features, respectively, with $s$ and $l$ denoting the spatial and language components, and csim represents the cosine-similarity function. Although the Frobenius-norm term is not strictly required in the loss function, we retain it to improve the numerical stability of the cosine-similarity term, which is undefined for vectors of zero norm.

**Visualizing the Distilled Semantic Features.** We examine the content of the distilled semantic features extracted by the vision backbones via principal component analysis (PCA), resulting in three-dimensional features which we visualize as images.

To quantify the geometric content of the distilled semantic features, we introduce the *geometric fidelity factor* (GFF), which captures the edge information present in the semantic features relative to the physical scene, as determined by the RGB image. To do so, we apply the Sobel–Feldman operator (Duda & Hart, 1973) to the RGB image. We apply hard-thresholding to the norm of the gradients to produce edges at varying resolutions. We post-process the semantic image $I_v$ and RGB image to obtain the binary edge masks $I_{e,\mathrm{sem}} \in \mathbb{R}^{W \times H \times 3}$ and $I_{e,\mathrm{rgb}} \in \mathbb{R}^{W \times H \times 3}$, respectively. After extracting edges from the semantic and RGB images, we compute the GFF using:

$$\mathrm{GFF} := \sum_{(i,j,k)} I_{e,\mathrm{sem}}[i,j,k] / \sum_{(i,j,k)} I_{e,\mathrm{rgb}}[i,j,k], \tag{1}$$

representing the fraction of edges retained by the distilled features. We examine the relative geometric content of the semantic features at different gradient thresholds in Section 5.

**Semantic Localization.** To further validate our conclusions on vision encoder features, we consider the downstream task of semantic localization. The distilled semantic features enable open-vocabulary object localization within the radiance field given a query: e.g., "find me a mug." For semantic localization, we compute the semantic embeddings of the language query $\phi_{\mathrm{query}}$ using CLIP and subsequently compute the cosine similarity between the query and all points in the radiance field to identify candidate matches.

**Inverting Radiance Fields.** We further explore the pretrained vision features in the inversion of radiance fields. Whereas the forward problem of image rendering in radiance fields is well-posed, the inverse problem is particularly challenging, especially without any simplifying assumptions such as those that assuming access to a good initial guess. In fact, the requirement of a good initial guess constitutes arguably the most significant challenge facing existing methods (Yen-Chen et al., 2021; Chen et al., 2025). Although radiance field inversion is related to a number of other problems, e.g., camera relocalization (Kendall et al., 2015; Xue et al., 2020; Zhou et al., 2024) and 3D scene reconstruction (Leroy et al., 2024; Wang et al., 2025), a few key characteristics distinguish radiance field inversion from these tasks. While camera relocalization methods generally require lots of training data (on the order of thousands to tens of thousands), radiance field inversion typically utilizes training data with fewer than a hundred (or a few hundred) samples, which makes camera relocalization methods impractical in this setting. Likewise, radiance field inversion methods computes poses in a global reference frame, unlike 3D scene reconstruction methods. As a result, we introduce

SPINE, a novel algorithm for inverting radiance fields using distilled semantics for camera pose recovery without an initial guess with small datasets. Moreover, SPINE enables us to comprehensively evaluate the relative performance of different vision features in radiance field inversion problems. SPINE learns a neural field $p_\psi : \mathbb{R}^d \mapsto \mathcal{P}$ which maps semantic (image) embeddings $f(\mathcal{I}) \in \mathbb{R}^d$ to a distribution over candidate poses, where $\mathcal{P}$ denotes the space of valid distributions. For DINO, we use the class token as input to $p_\psi$; for VGGT, we use the camera embeddings; for DUSt3R, MASt3R, and MUSt3R, we use the average image embedding as the input.

We decompose the camera pose $P \in \mathrm{SE}(3)$ into its translation $\mathbf{t} \in \mathbb{R}^3$ and orientation $\mathbf{R} \in \mathrm{SO}(3)$. Note that optimizing over the space of orientations is non-trivial, given that the orthogonality constraint in $\mathrm{SO}(3)$. To circumvent this challenge, SPINE parameterizes the camera orientation using the corresponding Lie algebra $\mathfrak{so}(3)$, the vector space of three-dimensional skew-symmetric matrices. Leveraging the isomorphism between $\mathfrak{so}(3)$ and $\mathbb{R}^3$, we represent the camera rotation by $\mathbf{r} \in \mathbb{R}^3$. Note that we can construct a skew-symmetric matrix from $\mathbf{r}$ and subsequently map elements of $\mathfrak{so}(3)$ to $\mathrm{SO}(3)$ using the exponential map, i.e., $\exp : \mathfrak{so}(3) \mapsto \mathrm{SO}(3)$. Moreover, we make no additional assumptions beyond those made by the underlying radiance field and train SPINE entirely on the same inputs as the radiance field using the mean-squared-error (MSE). Like other radiance field inversion and camera relocalization methods, SPINE is trained per scene to estimate poses in an absolute (inertial) frame. We describe further implementation details in Appendix A.2

## 4 PROBING PRETRAINED SEMANTICS

Here, we probe the semantic features from pretrained vision encoders to deconstruct the image embeddings extracted by these encoders. We train distilled radiance fields in nine scenes from three widely-used benchmark datasets, namely: *Ramen*, *Teatime*, and *Waldo_kitchen* in the LERF dataset (Kerr et al., 2023); *Bed*, *Covered Desk*, and *Table* in the 3D-OVS dataset (Liu et al., 2023); and *Office*, *Kitchen*, and *Drone* in the Robotics dataset (Shorinwa et al., 2025). We note that the 3D-OVS dataset primarily contains small-scale scenes while the LERF and Robotics datasets contain larger-scale scenes. Unlike the LERF dataset which contains mostly curated scenes, the Robotics dataset contains more in-the-wild scenes. For all scenes, we learn both GS and NeRF representations, from which we generate synthetic images rendered at novel viewpoints, augmented with distilled semantic embeddings. We analyze these semantic embeddings to identify the image features that the vision encoders attend to and provide visualizations of these *attention* maps for better understanding. We discuss additional implementation details in Appendix A.3.

### 4.1 SEMANTIC CONTENT IN SMALL-SCALE BENCHMARK SCENES

We visualize the contents of the semantic features of DINOv2, DINOv3, DUSt3R, MASt3R, MUSt3R, and VGGT rendered in the 3D-OVS scenes to characterize the information prioritized by these encoders in Figure 3. We emphasize that in general, these pretrained encoders were not specifically trained on the selected scenes from the specific datasets. Nonetheless, the learned representations from these models generalize effectively to the selected scenes, as observed in the figure. First, we highlight that the visual-semantic encoders (DINOv2/DINOv3) attend to entity-level semantic information that resolve the identities of different objects. For example, the embeddings for the person's hand, camera, bag, and shoe lie within very distinct areas in the semantic space, as indicated by the unique colors assigned to each object. However, the boundaries of these objects are not precisely defined in the semantic space, indicating a lack of focus on extracting precise geometry. In contrast, the semantic features of the geometry-grounded encoders (DUSt3R, MASt3R, MUSt3R) contain a remarkable level of structural detail, highlighting the checkered (striped) pattern on the bedsheet and the cross-hatched pattern on the bag. These features still contain some object-level semantic information but at a much lower resolution compared to the visual-semantic features. Similarly, VGGT prioritizes structural information—essentially tracing the outline of the objects—while retaining less object-level semantic information.

We further analyze the content of these semantic features through the GFF, defined in Equation (1). In line with the visualizations in Figure 3, we observe that geometry-supervised encoders extract features that contain more spatial information, especially at lower thresholds in the GFF, shown in the bottom panel of Figure 4. At larger thresholds, some of the low-intensity edges inherent in these features are filtered out, leading to drop in the GFF, especially with DUSt3R and MASt3R.

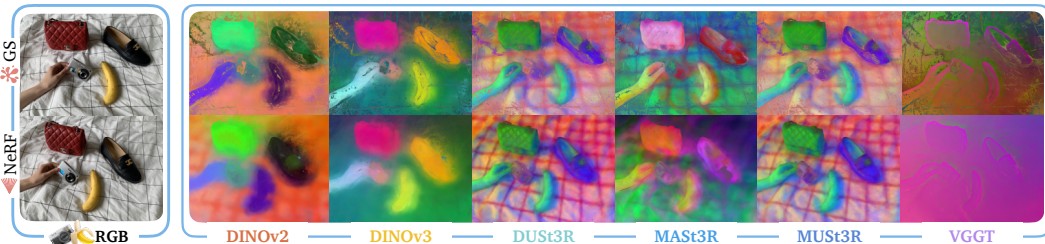

Figure 3: **Semantic content of distilled features in 3D-OVS (*Bed*).** Whereas visual-semantic features mostly capture object-level semantic information, geometry-grounded features prioritize more structural details, such as an object's outline, at the expense of object-level semantics.

Consequently, all methods achieve almost the average GFF, except DUSt3R and MASt3R. These findings can be explained by the relative simplicity of the scene with a notable lack of diversity in objects. We hypothesize that greater diversity could broaden the gap in the GFF achieved by each method, which we explore in the rest of this section. We also provide additional results in NeRFs in Appendix A.4.

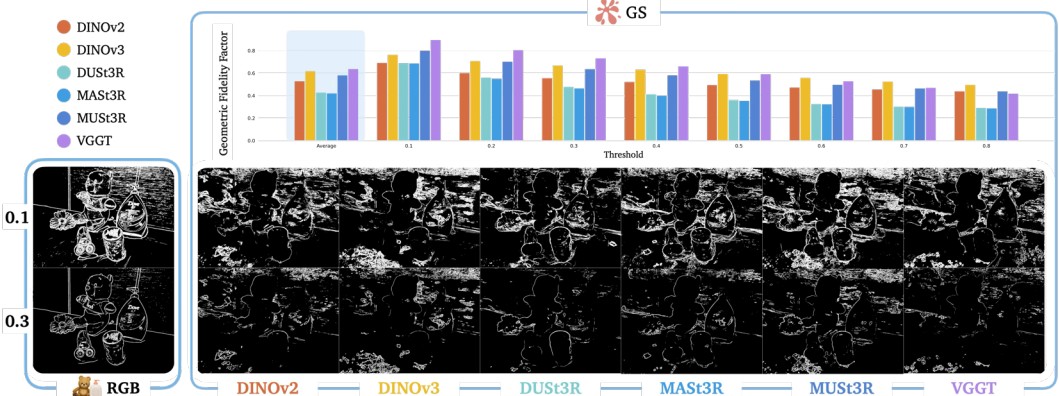

Figure 4: **Geometric fidelity factor (GFF) in 3D-OVS.** Geometry-grounded features prioritize spatial detail, e.g., object edges compared to visual-semantic features.

## 4.2 SEMANTIC CONTENT IN CURATED SCENES (LERF)

Here, we examine the content of the semantic features from the vision encoders in the LERF dataset. In Figure 5, we see that geometry-grounded encoders prioritize structural information, as demonstrated by the well-defined outlines of the table and chair. Qualitatively, all semantic encoders except VGGT retain similar amounts of object-level semantic information. These results further underscore that geometry-grounded encoders are more focused on preserving spatial information. Furthermore, we explore the structural content of these features in Figure 6. Similar to the results in the 3D-OVS scenes, we find that geometry-grounded encoders capture the morphology of objects better than DINO, particularly at lower thresholds. For example, in the grayscale images in the bottom panel of Figure 6, we can easily identify the profile of objects on the table when using the spatially-grounded encoders compared to DINO.

## 4.3 SEMANTIC CONTENT FOR IN-THE-WILD SCENES

Now, we consider in-the-wild scenes containing diverse objects in different configurations. In Figure 7, we see that the spatially-supervised encoders capture more information on the geometry of the scene, with prominent edges defining the outline of the objects in the scene. In particular, VGGT attends strongly to the object edges compared to other elements of the scene. The strong attention of the models to the objects' geometry is supported by their GFF scores in Figure 8. Across all thresholds,

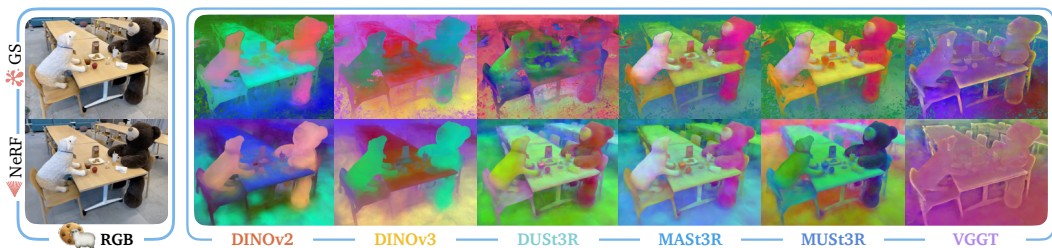

Figure 5: **Semantic content of distilled features in LERF (*Teatime*).** Spatially-grounded encoders capture the morphology of objects better than DINO, indicating their propensity to attend to structural information.

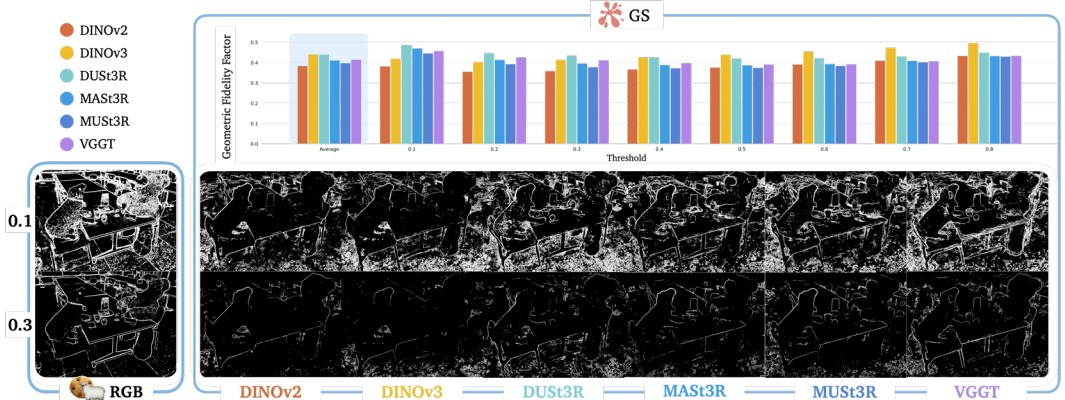

Figure 6: **Geometric fidelity factor (GFF) in the LERF Scene.** VGGT's features contain prominent object edges, unlike visual-only semantic features.

the semantic features of these encoders contain more structural details compared to visual-semantic encoders, as visualized in the bottom panel. Note that the difference in the GFF scores is more noticeable due to the density and diversity of objects in these scenes compared to the other scenes.

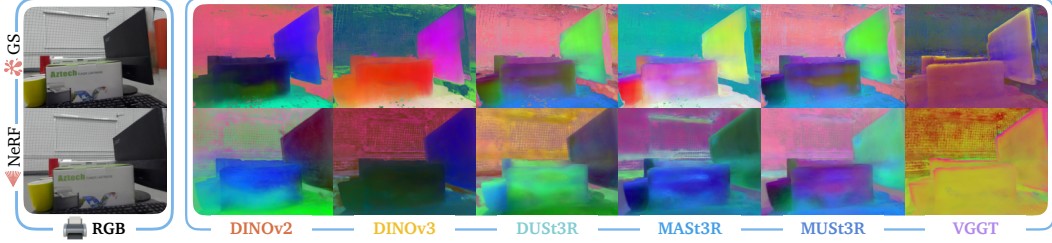

Figure 7: **Semantic content of distilled features in the Robotics dataset (*Quadruped Office*).** Whereas visual-semantic features provide object-level information, geometry-grounded features provide more structural details, such as an object's contour.

## 4.4 SUMMARY

Across all datasets, we observe that visual-semantic features (e.g., DINO) capture higher-resolution object-level semantic detail; however, these features do not encapsulate high-fidelity object morphology. In contrast, geometry-supervised encoders give up entity-level semantic information to attend to structural components of the scene. Importantly, our findings suggest that more significant finetuning generally leads to a greater shift in the attention of these pretrained encoders from object-level semantic information to the objects' geometries. Further, these results suggest that visual-semantic

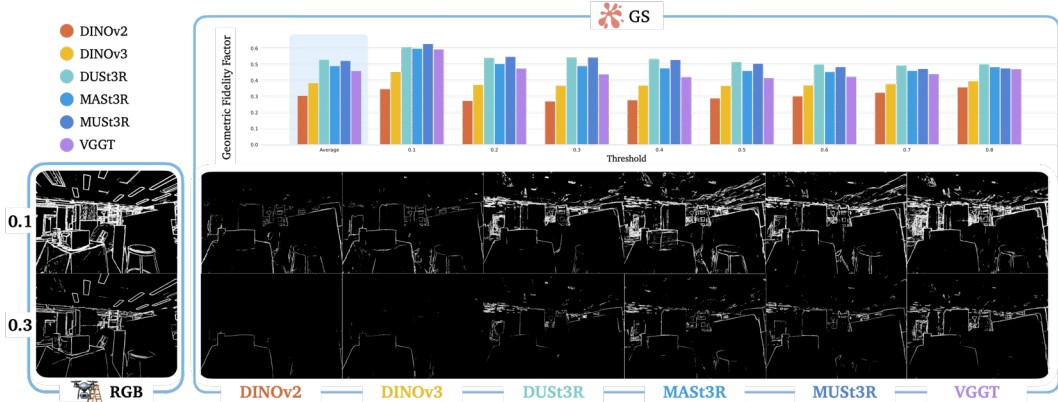

Figure 8: **Geometric fidelity factor (GFF) in the Robotics Scenes.** VGGT's features contain prominent object edges, unlike visual-semantic features.

encoders would excel at tasks that do not require knowledge of the precise geometry of objects, like many pick-and-place tasks in robot manipulation; however, dexterous manipulation tasks, such as multi-finger prehensile tasks, might benefit from task-supervised vision encoders, such as the geometry-grounded encoders discussed in our work. We leave an exhaustive examination of these potential applications to future work.

In addition, our findings provide insight into contradictory claims made by prior work. Specifically, some existing papers (Chi et al., 2025) claim that pretrained task-specific encoders outperform visual-semantic encoders (e.g., CLIP) in robot manipulation tasks, while others (Huang et al., 2025) claim that the opposite is true. Our findings clarify that the observed performance of these encoders strongly depends on the evaluation task. For example, the work in (Chi et al., 2025) studies more complex manipulation tasks where object geometry is more important compared to the tasks in (Huang et al., 2025). Likewise, our findings suggest that finetuning vision encoders could degrade their generalization by significantly changing their attention maps, providing insights into the results in (Karamcheti et al., 2024).

We can attribute the degradation in the semantic knowledge of the finetuned encoders to catastrophic forgetting, where the encoders lose prior knowledge of object-level semantic information as finetuning progresses. As demonstrated in prior work, our findings suggest that finetuning with low-rank adaptation (LoRA) could be essential in preserving the semantic knowledge of vision encoders. Further, finetuned encoders tend to overfit to the task, which can be addressed through multi-objective training frameworks using loss functions with a task-focused component and a task-agnostic component to promote generalization.

## 5 EXPERIMENTS

We validate our findings on the content of pretrained semantic features, using distilled radiance fields as a testbed to enable fine control over confounding variables that could otherwise hurt the interpretability of our results. We consider two main robotics tasks: semantic object localization and radiance field inversion. We conduct extensive experiments across nine scenes from three benchmark datasets, evaluating the performance of the semantic features from each vision encoder. We present additional details of the evaluation setup and results in the Appendix.

### 5.1 SEMANTIC OBJECT LOCALIZATION

We examine the performance of the semantic features from all encoders in semantic object localization using the procedure described in Appendix A.2. In each scene, we use CLIP to encode the natural-language queries and subsequently generate the continuous relevancy mask. We use GroundingDINO (Liu et al., 2024) and SAM-2 (Ravi et al., 2024) to annotate the ground-truth segmentation mask, used in computing the segmentation accuracy metrics: SSIM, PSNR, and LPIPS. Figure 9 summarizes

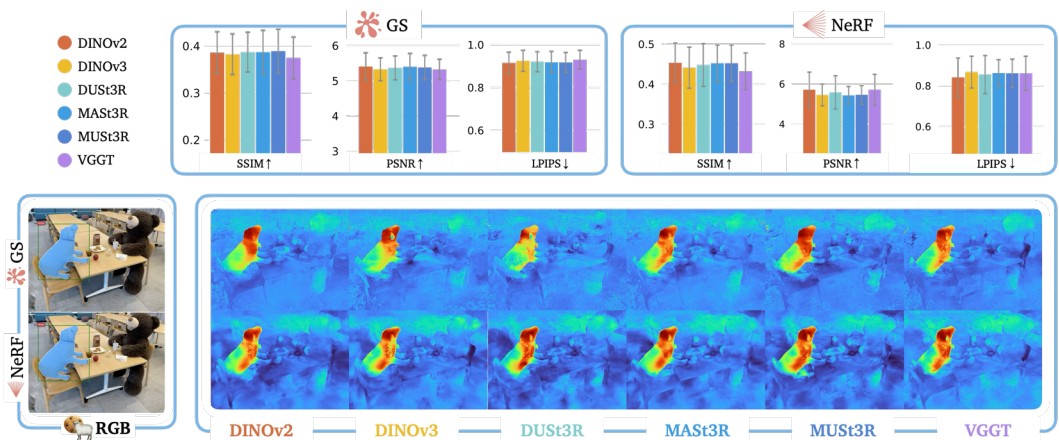

Figure 9: **Semantic object localization.** Both visual-semantic features and geometry-grounded features achieve similar localization accuracies (*Teatime* scene visuals).

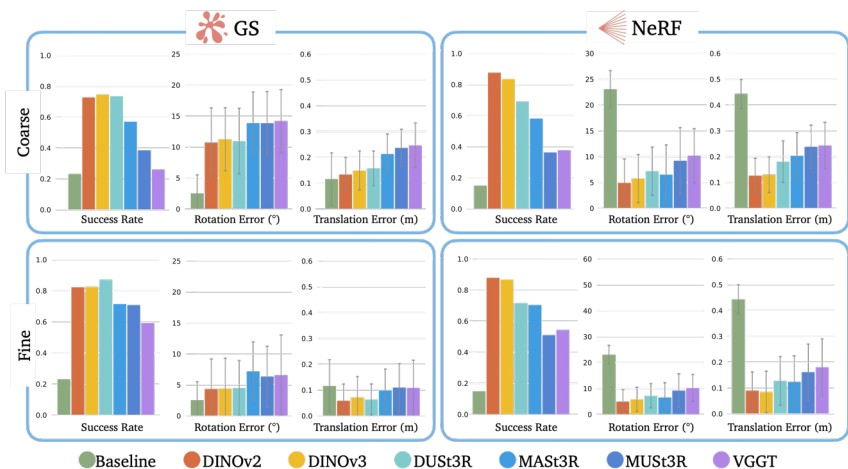

Figure 10: **Inverting radiance field.** Even with only coarse inversion, SPINE outperforms the baseline methods in success rate (GS and NeRF) and accuracy (NeRF) without an initial guess. We emphasize that the seemingly high accuracy of the GS baseline is primarily due to the low success rate. The fine inversion step significantly improves both the success rate and accuracy of SPINE.

our results. We find no significant difference in the localization accuracy achieved by the different semantic features across GS and NeRF, suggesting that both semantic features are effective in co-supervising CLIP for open-vocabulary localization. However, we observe marginal degradation in performance with the task-specific (geometry-grounded) features, in line with the expectations in Section 4. In addition, we visualize the ground-truth RGB and segmentation mask and the relevancy masks in the *Teatime* scene, highlighting the effectiveness of both kinds of semantic features in localizing the cookies, sheep, and bear. We provide additional results in Appendix A.5.

## 5.2 INVERTING RADIANCE FIELDS

We evaluate the accuracy of the semantic features from the vision encoders in radiance field inversion across all benchmark scenes. For the baseline methods Splat-Loc (Chen et al., 2025) and iNeRF (Yen-Chen et al., 2021), we use an initial guess with rotation and translation errors of $R_{err} = 30 \deg$ and $T_{err} = 0.5$m, respectively. We define the success rate based on a threshold on the rotation and translation error, further described in Appendix A.3. In Figure 10, we provide the results on the accuracy and success rate achieved by each encoder, averaged across all the scenes. We find that the

visual-semantic features (e.g., DINO) generally outperform geometry-grounded features. Although this finding might seem surprising at first glance, the results are actually well-aligned with intuition. Specifically, SPINE relies significantly on object-level semantics for global pose recovery. However, the geometry-grounded encoders discard this semantic information to attend to structural content, especially when more extensively finetuned, e.g., MUSt3R/VGGT. As a result, these models often fail to outperform the other encoders. In contrast, DUSt3R remains competitive with the visual-semantic encoders, which can be explained by its relatively lightweight finetuning procedure.

Considering only the coarse inversion phase, SPINE achieves significantly higher success rates compared to the baselines in both GS and NeRF scenes, which is not surprising. Essentially, all prior methods struggle to return an accurate solution without a good initial guess, a major challenge faced by these methods. SPINE increases the absolute success rates by about $60\%$ with DINO features. Further, we observe that SPINE achieves higher accuracy in all NeRF scenes and performs competitively in the GS scenes. We emphasize that the seemingly higher accuracy of the GS baseline is primarily due to its low success rate (which effectively represents its accuracy on relatively easy problem). SPINE computes sufficiently accurate solutions in more challenging problems. We also observe a slight degradation in accuracy in geometry-grounded encoders.

Further, we observe the importance of the fine inversion phase in boosting both the success rate and accuracy of SPINE. Concretely, we see improvements in the success rates after the fine inversion procedure, ranging from about $5\%$ to about $35\%$ depending on the initial success rate after coarse inversion. More noteworthily, fine inversion results in a decrease in the rotation and translation error. In GS, the fine estimates effectively match or surpass the accuracy achieved by the baseline when the baseline succeeds and consistently outperform the baseline in the case of NeRFs. SPINE offers competitive runtimes compared to the baselines. After the one-time setup pass, SPINE runs at about 2 Hz, essentially as fast as the method in (Chen et al., 2025) and much faster than iNeRF, which runs at 0.05 Hz for 100 optimization steps. We provide additional results for each scene in Appendix A.6.

## 6 CONCLUSION

We explore the content of semantic features of robotics vision encoders to better understand their characteristics as a guide to more effectively integrating these vision encoders into robotics pipelines, such as foundation models. Our studies reveal that visual-semantic encoders show a propensity to focus on object/part-level semantic information that is important for distinguishing between different classes of objects, while geometry-grounded encoders prioritize structural information that emphasize fine-grained edges and other spatial details over object-level semantic information. Further, our work reconciles contradictory claims on vision encoders made by prior work, relating to their performance in robotics and the effect of finetuning them. To enable the evaluations in our work, we derive a novel method for inverting radiance fields and demonstrate its superior performance compared to existing baselines, without requiring an initial guess, unlike the baselines.

## 7 LIMITATIONS AND FUTURE WORK

**Self-Supervised Geometry-Grounding.** Our findings suggest that existing solutions for geometry-grounding may lead to catastrophic forgetting of pretrained features. Future work will explore self-supervised approaches for spatial-grounding to mitigate catastrophic forgetting, improve adaptability, and enable larger-scale pretraining.

**Synergy between Geometry and Vision.** In addition, our experiments revealed that geometry-grounded semantics did not improve the semantic object localization accuracy, despite its more significant structural content, likely due to the loss of valuable object-level semantic information. Hence, these features fail to realize the synergy between geometry and vision. Future work will introduce more effective strategies for establishing synergy between the geometry-grounded and visual-semantic features for more robust scene understanding.

**Efficient Inference.** Existing geometry-grounded vision backbones require notable compute overhead compared to ungrounded backbones, amplified by the absence of lightweight variants. Future work will examine more efficient architectures for spatially-grounded vision backbones to enable their use in real-time applications, e.g., in robot manipulation.

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
