# A APPENDIX

## A.1 PRELIMINARIES

We discuss semantic distillation in radiance fields.

**NeRFs.** NeRFs learn implicit volumetric color and density fields, encoding the occupancy and radiance of each point in the scene, given a set of images $\mathcal{I}$ and corresponding camera poses, which is typically computed via structure-from-motion (Schonberger & Frahm, 2016). Specifically, the color field $\mathbf{c} : \mathbb{R}^3 \times \mathbb{S}^2 \mapsto \mathbb{R}^3$ maps a 3D point $\mathbf{x} \in \mathbb{R}^3$ and a camera viewing direction $d \in \mathbb{S}^2$ to an RGB color $\mathbf{c}(\mathbf{x}, d)$. Likewise, the density field $\rho : \mathbb{R}^3 \mapsto \mathbb{R}_+$ maps $\mathbf{x}$ to a non-negative volume density $\rho(\mathbf{x})$, representing the differential probability of a light ray terminating at a particle located at $\mathbf{x}$. Using ray-marching to render images from $\mathbf{c}$ and $\rho$, NeRFs utilize stochastic gradient descent to optimize the parameters of $\mathbf{c}$ and $\rho$ (represented as MLPs), minimizing the photometric error between the rendered and ground-truth images.

**Gaussian Splatting.** Gaussian Splatting (GS) utilizes 2D (Huang et al., 2024) or 3D (Kerbl et al., 2023) Gaussian primitives to represent non-empty space, each defined by a mean $\mu \in \mathbb{R}^3$, covariance $\Sigma \in \mathbb{S}_{++}^3$, opacity $\alpha \in \mathbb{R}_+$, and spherical harmonics (for view-dependent color) parameters. Like NeRFs, GS optimizes these parameters by minimizing the photometric error between rendered and ground-truth images, initialized from a sparse point cloud generally computed using structure-from-motion. Notably, GS employs a tile-based rasterization procedure to efficiently project the Gaussian primitives to the image plane, circumventing the expensive volumetric ray-marching procedure used by NeRFs, for faster training and rendering. Moreover, the explicit scene representation enables relatively easier theoretical analysis compared to NeRFs, in addition to providing more accurate depth estimation and mesh extraction.

**Semantic Distillation.** Prior work (Zhi et al., 2021; Kobayashi et al., 2022) has explored distilling semantic information from foundation models into NeRFs and GS for open-vocabulary semantic segmentation, object localization, and scene-editing. We limit our discussion to distillation methods that learn a neural field (Kerr et al., 2023), due to their generality to both NeRFs and GS and faster training/inference speeds (Shorinwa et al., 2024a). Neural semantic distillation methods train a semantic field $f : \mathbb{R}^3 \mapsto \mathbb{R}^d$ mapping a 3D point to a $d$-dimensional semantic embedding. In general, these methods leverage hashgrid encodings (Müller et al., 2022) to learn varying resolutions of semantic detail. The semantic field is trained simultaneously with the radiance field with direct supervision by the ground-truth embeddings from the teacher network, e.g., CLIP/DINO.

## A.2 IMPLEMENTATION DETAILS

**Extracting Pretrained geometry-grounded Semantic Features.** For VGGT, we extract ground-truth pretrained semantic embeddings for each image from the depth and point heads of VGGT and its intermediate layers, which were trained for depth estimation and dense point cloud reconstruction, respectively.

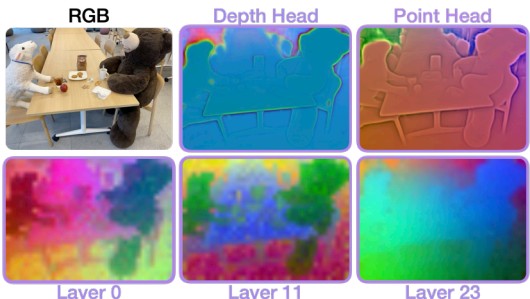

We visualize these semantic embeddings in Figure 11 using the first three principal components. We find that the point head produces features with the highest-fidelity spatial detail, compared to the depth head and other intermediate layers. For all other vision encoders, we follow the standard procedure to extract ground-truth features.

Figure 11: VGGT's semantic embeddings from different heads, showing high-fidelity geometric content of the point head.

**Visualizing the Distilled Semantic Features.** We perform PCA to examine the distilled feature contents, producing three-dimensional features which we visualize as images. Concretely, given a matrix $\hat{I}_f \in \mathbb{R}^{m \times d}$ of $d$-dimensional semantic features, we compute the singular value decomposition (SVD) of $A$ to obtain the tuple $(U, S, V)$,

such that: $\hat{I}_f = U\Sigma V^T$, where $U \in \mathbb{R}^{m \times p}$, $V \in \mathbb{R}^{n \times p}$, and $\Sigma \in \mathbb{R}^{p \times p}$ denotes the left , right , and singular values, respectively. In practice, we compute the low-rank SVD for computational efficiency. Subsequently, we map the semantic features in $I_v$ to the RGB image space using the first-three principal components via the transformation: $I_v = \hat{I}_f V_{[:,:3]}$, with $I_v \in \mathbb{R}^{m \times 3}$, which is reshaped into a 3-channel 2D image for visualization (i.e., $I_v \in \mathbb{R}^{W \times H \times 3}$).

**Semantic Localization.** For increased robustness, we utilize the semantic relevancy score (Kerr et al., 2023) given by:

$$\nu(\phi_{\text{query}}) = \min_i \frac{\exp(\phi_{\text{query}} \cdot \phi_f)}{\exp(\phi_f \cdot \phi_{\text{canon}}^i) + \exp(\phi_{\text{query}} \cdot \phi_f)}, \tag{2}$$

where $\phi_f$ represents the rendered semantic embeddings from the radiance field and $\phi_{\text{canon}}$ represents the semantic embedding of *canonical* prompts, i.e., generic or negative prompts to better distinguish between confident localization matches from non-confident ones. We use canonical prompts such as: "object," "stuff," and "things." The semantic relevancy score can be viewed as the pairwise softmax over a set of positive and negative queries with respect to the rendered semantic embeddings. For conservative results, we take the minimum over all pairwise softmax distributions to define the semantic relevancy score.

To evaluate semantic localization accuracy, we map the ground-truth segmentation mask to the Euclidean space, assigning the value zero to pixels outside the map and one otherwise. Likewise, we normalize the semantic relevancy score generated by the radiance field at each view to values in $[0, 1]$ to obtain a relevancy mask. Afterwards, we map the ground-truth mask and the relevancy mask to the RGB space using a colormap. Thereafter, we compute the objective localization accuracy using widely-used perceptual metrics, such as the structural similarity index measure (SSIM), learned perceptual image patch similarity (LPIPS), and peak-signal-to-noise ratio (PSNR). We do not primarily use metrics that require binary masks, e.g., mean intersection-over-union (mIoU), as these metrics would require the selection of a similarity threshold for each rendered semantic relevancy image. Identifying an optimal similarity threshold for the vision backbone features individually presents computational challenges, especially since the optimal values are generally view-dependent. Moreover, approximating the optimal thresholds could lead to confounding results. Nonetheless, we include mIoU results computed with a threshold of $0.9$ for completeness.

**Camera pose estimation.** Given a query image, we compute the semantic embedding of the image, which is mapped to $\mathcal{P}$ using $p_\psi$. In general, the estimated camera pose is not always of sufficiently high accuracy. Consequently, we render images from the radiance field at the estimated coarse camera pose, using novel-view synthesis to generate an RGB-D image. Subsequently, we match image features from the query image to the rendered image, associated to a point-cloud in the radiance field through the rendered depth. Given the set of corresponding matches $\mathcal{C}$, we solve the perspective-$n$-point (PnP) problem to refine the coarse pose estimates: $(\hat{\mathbf{t}}, \hat{\mathbf{R}}) = \arg\min_{\mathbf{t} \in \mathbb{R}, \mathbf{R} \in SO(3)} \sum_{(\mathbf{p},\mathbf{q}) \in \mathcal{C}} \|\mathbf{p} - \mathbf{K}(\mathbf{R}\mathbf{q} + \mathbf{t})\|_2^2$, where $(\mathbf{p}, \mathbf{q})$ denote a corresponding pair with homogeneous image coordinates $p$ (in the query image) and 3D point $q$ (in the radiance field), $\mathbf{K}$ denotes the camera intrinsic matrix, and $\hat{\mathbf{t}}$ and $\hat{\mathbf{R}}$ represent the estimated camera translation and rotation, respectively. For robustness to spurious correspondences, we solve the PnP problem using RANSAC, mitigating the effects of outliers on the estimated pose.

We evaluate the accuracy of the pose estimates using the $SE(3)$ error, composed of the translation and rotation error between the ground-truth and estimated camera poses. We compute the translation error directly as the $\ell_2$ norm of the difference between the ground-truth and estimated translation and leverage the trace property of rotation matrices: $\text{trace}(\mathbf{R}) = 1 + 2\cos(\theta)$ to compute the smallest rotation angle required to align the ground-truth and estimated camera orientations, where $\theta$ denotes the angle associated with rotation matrix $\mathbf{R}_\delta$. Specifically, the absolute rotation error $\theta$ is given by: $\theta = \arccos\left(\frac{\text{trace}(\mathbf{R}_\delta) - 1}{2}\right)$, where $\mathbf{R}_\delta = \hat{\mathbf{R}}^\top \mathbf{R}_{\text{gt}}$ denotes the relative rotation matrix between the the ground-truth and estimated rotation matrices, $\hat{\mathbf{R}}$ and $\mathbf{R}_{\text{gt}}$, respectively.

## A.3 EVALUATION SETUP

**Implementation.** We train a GS and NeRF representation per scene using the camera poses provided with the datasets for 30000 iterations within the Nerfstudio framework (Tancik et al., 2023). To

extract geometry-grounded semantics, we use the official VGGT checkpoints provided by the authors of (Wang et al., 2025). Likewise, we extract visual-semantic semantics from DINOv2 (Oquab et al., 2023) and DINOv3 (Siméoni et al., 2025) using their official checkpoints and use the CLIP ResNet model RN50x64 for language semantics. We parameterize the semantic fields $f_l$ and $f_s$ using a multi-layer perceptron (MLP) with two layers having 32 neurons per layer and ReLU activation. The shared hashgrid uses a two-layer MLP with 16 neurons per layer. We train the radiance fields on an Nvidia L40 GPU with 48GB VRAM, although we emphasize that in many cases, the GPU memory is not the main compute bottleneck in our implementation. In fact, preliminary experiments were conducted on an Nvidia GeForce RTX 4090 GPU with 24GB VRAM. For image feature extraction and matching in PnP, we use SIFT (Lowe, 2004). We train SPINE alongside the radiance field using the same training data. We parametrize the SPINE field using an two-layer MLP with 16 neurons per layer.

**Metrics.** We evaluate the relative performance between spatially-grounded and visual-semantic semantics using task-specific metrics. For semantic content analysis, we use the geometric fidelity factor (GFF) metric (see Section 3). For semantic object localization and radiance field inversion, we use the perceptual metrics SSIM, PSNR, and LPIPS (see Appendix A.2), and the success rate with respect to a scene-specific error threshold, as well as $SE(3)$ error with rotation in degrees and translation in meters (see Section 3), respectively. We compute these metrics across 100 novel (unseen) camera poses in most scenes. In the 3D-OVS dataset, we use the training camera poses due to the limited scene coverage.

## A.4 ADDITIONAL SEMANTIC CONTENT RESULTS

Below, we include visualizations for the PCA and segmentation experiments, sampling one or two camera views from each scene.

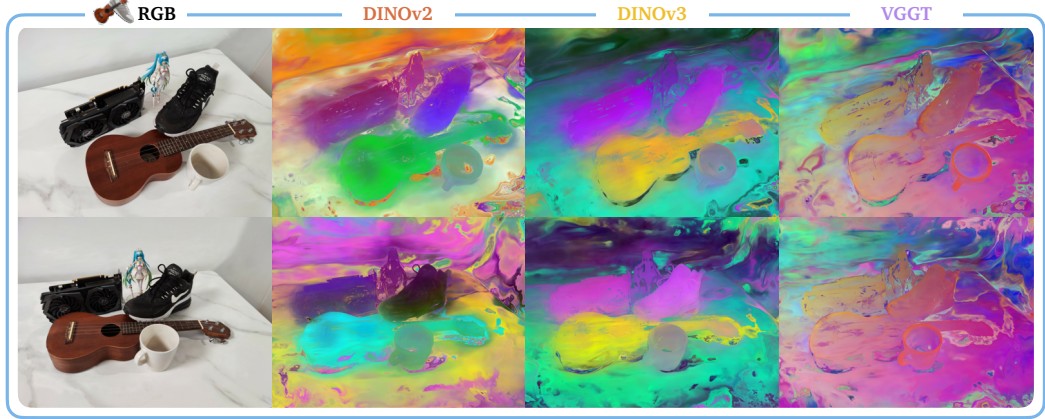

Figure 12: Semantic content of distilled features visualized for GS for the 3D-OVS table scene.

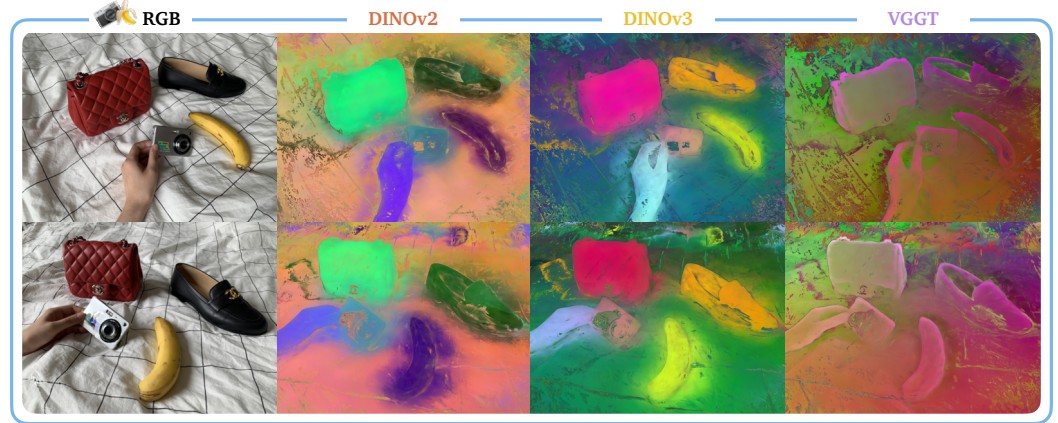

Figure 13: Semantic content of distilled features visualized for GS for the 3D-OVS bed scene.

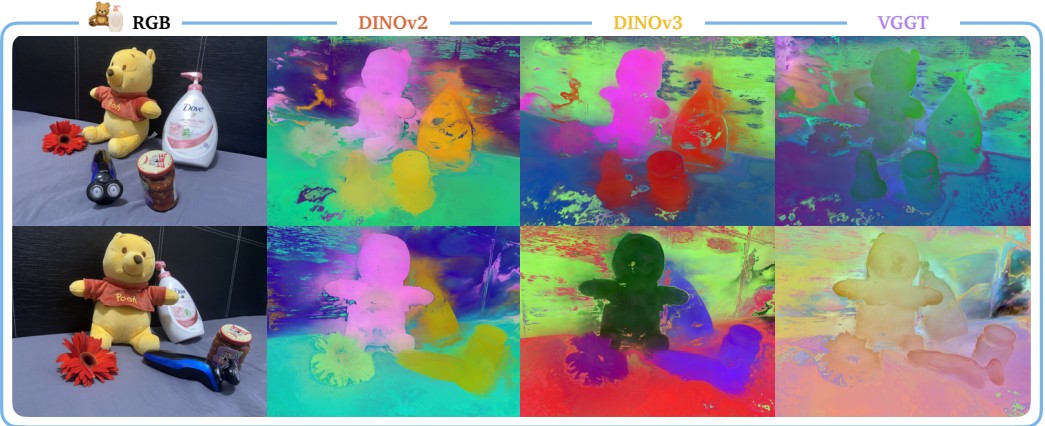

Figure 14: Semantic content of distilled features visualized for GS for the 3D-OVS covered desk scene.

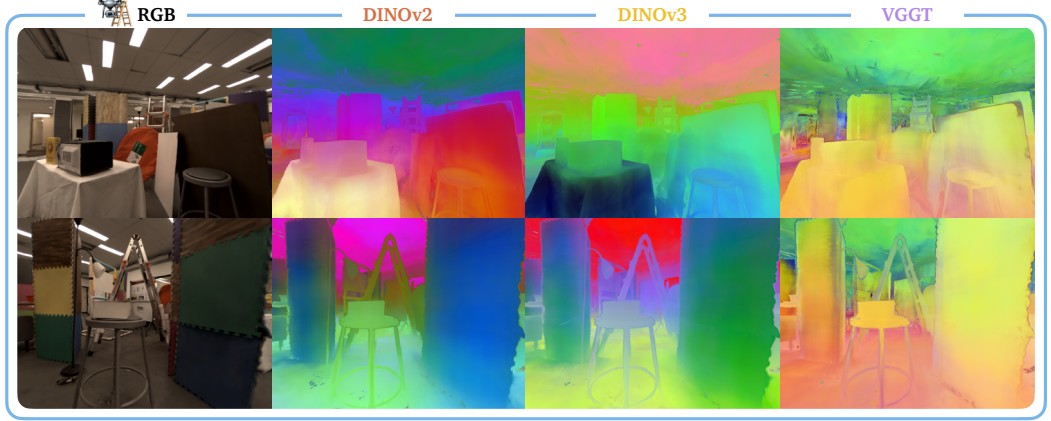

Figure 15: Semantic content of distilled features visualized for GS for the drone scene.

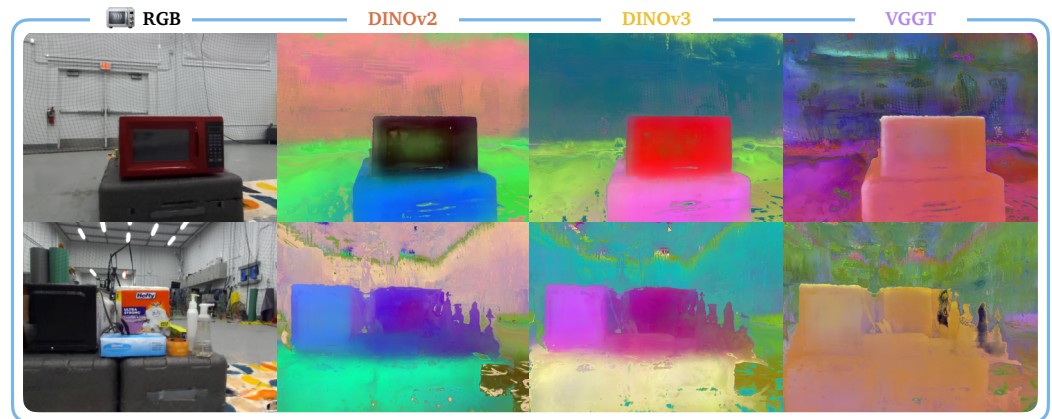

Figure 16: Semantic content of distilled features visualized for GS for the quadruped kitchen scene.

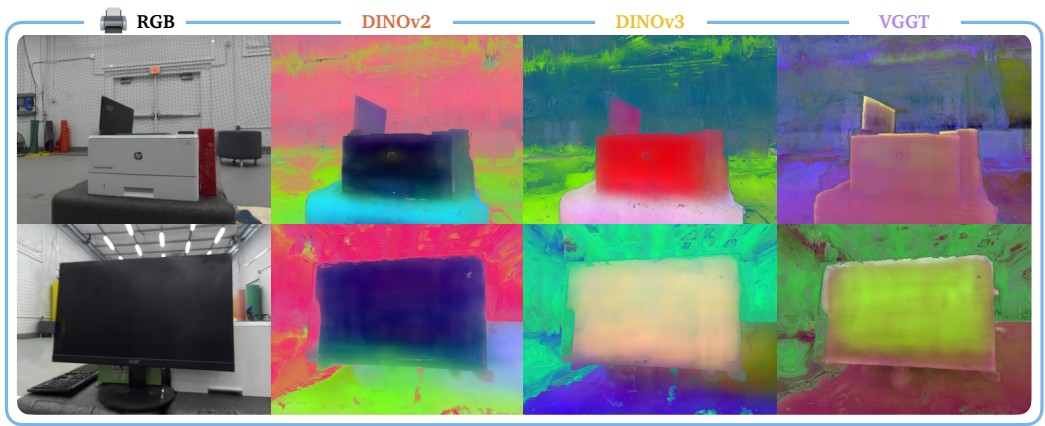

Figure 17: Semantic content of distilled features visualized for GS for the quadruped office scene.

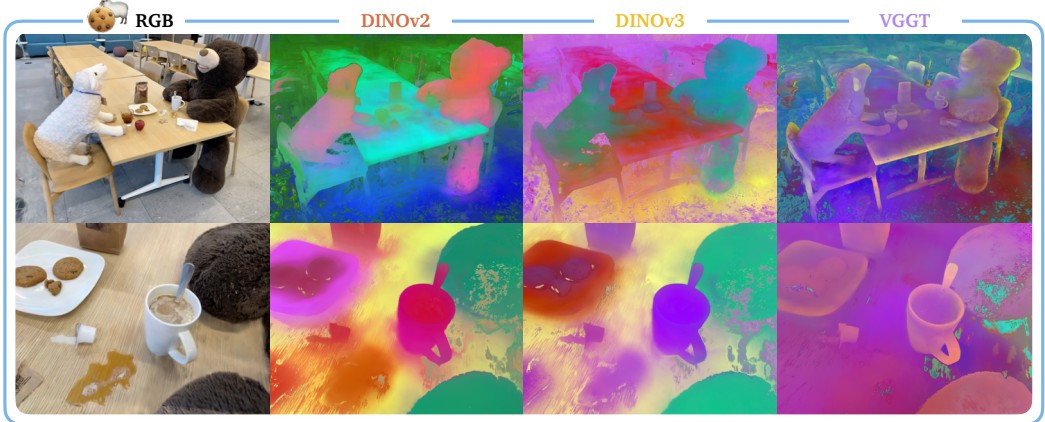

Figure 18: Semantic content of distilled features visualized for GS for the LERF teatime scene.

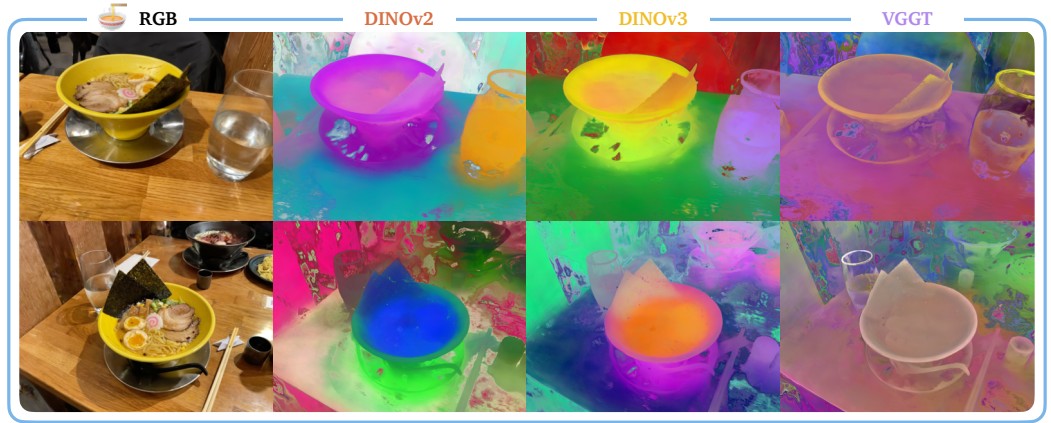

Figure 19: Semantic content of distilled features visualized for GS for the LERF ramen scene.

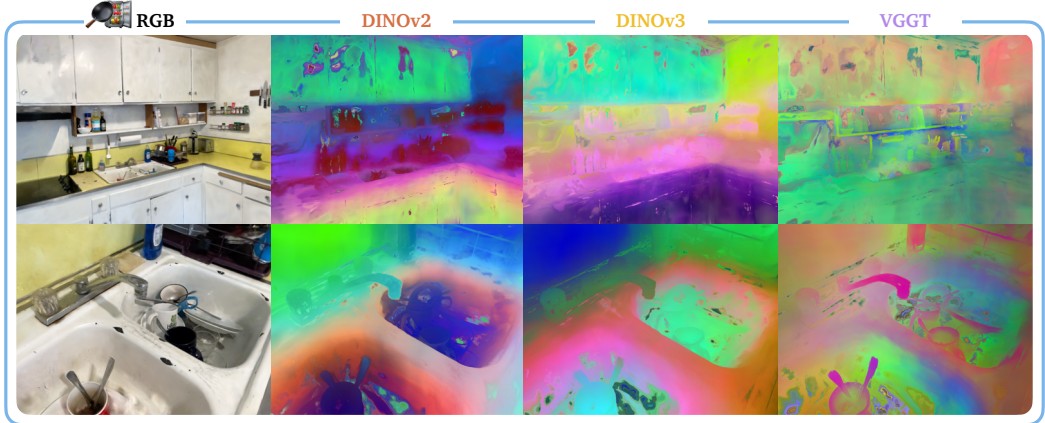

Figure 20: Semantic content of distilled features visualized for GS for the LERF kitchen scene.

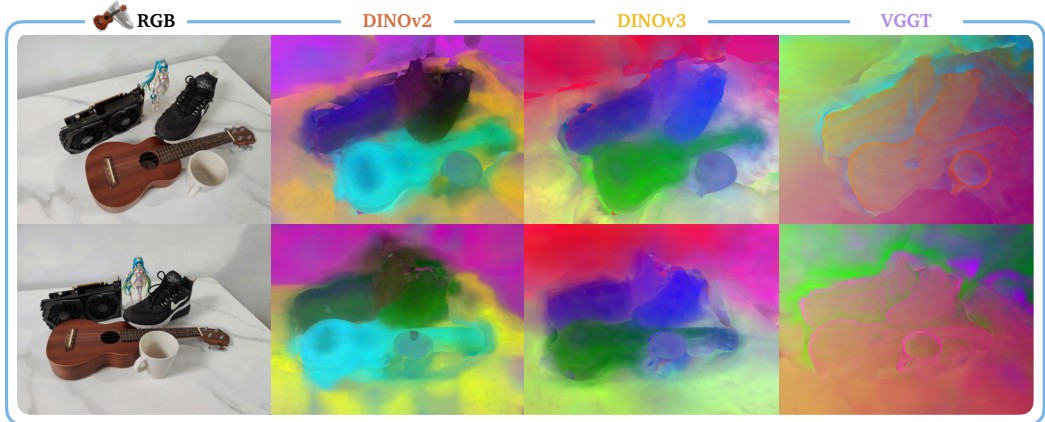

Figure 21: Semantic content of distilled features visualized for NERF for the 3D-OVS table scene.

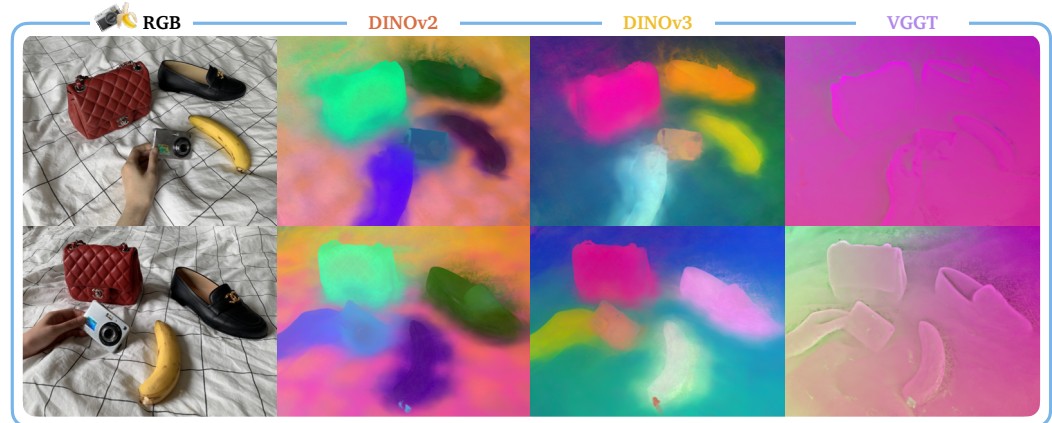

Figure 22: Semantic content of distilled features visualized for NERF for the 3D-OVS bed scene.

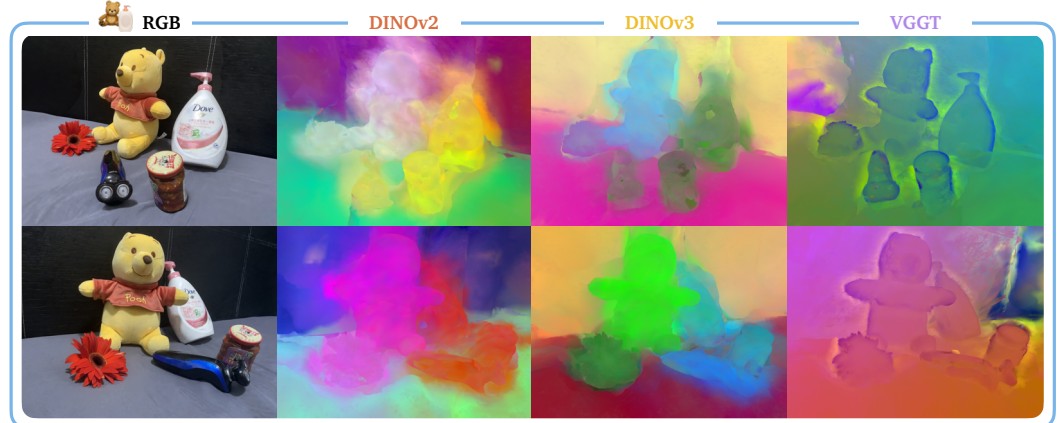

Figure 23: Semantic content of distilled features visualized for NERF for the 3D-OVS covered desk scene.

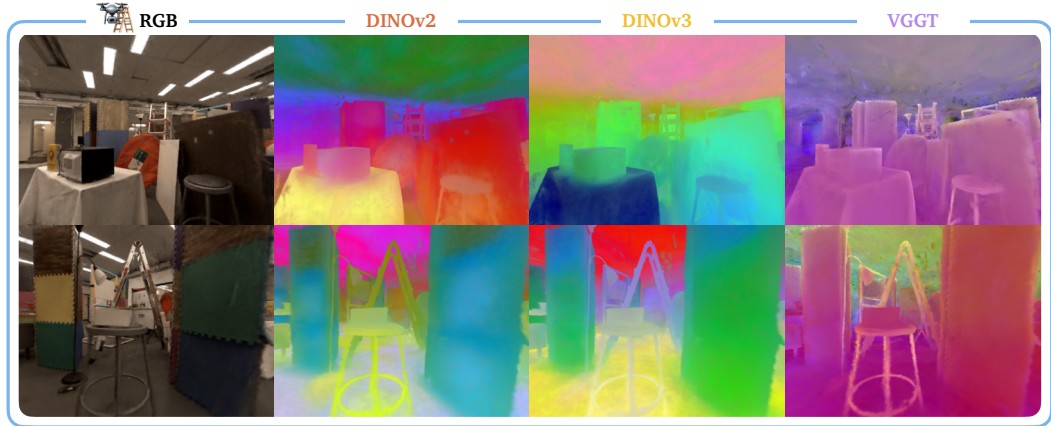

Figure 24: Semantic content of distilled features visualized for NERF for the drone scene.

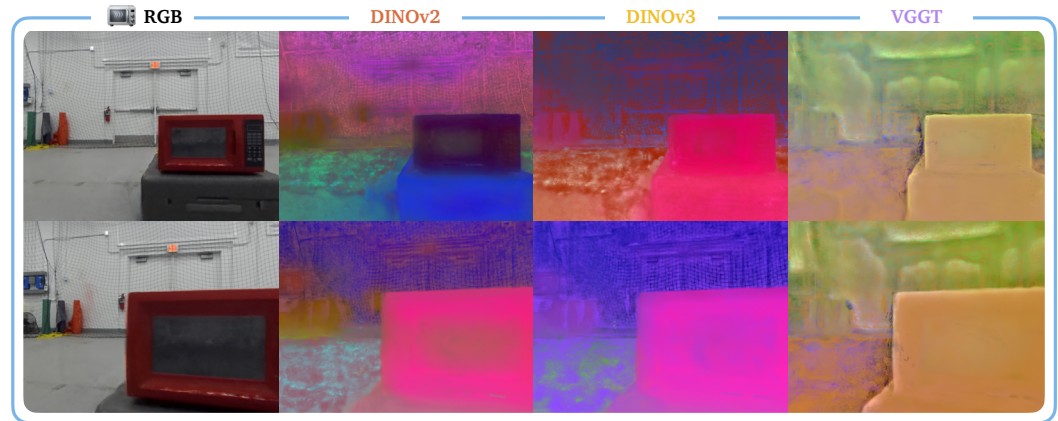

Figure 25: Semantic content of distilled features visualized for NERF for the quadruped kitchen scene.

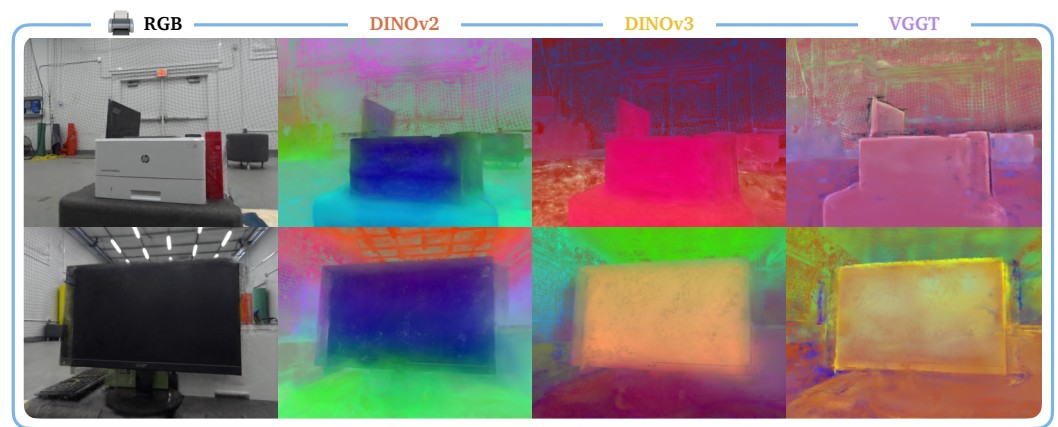

Figure 26: Semantic content of distilled features visualized for NERF for the quadruped office scene.

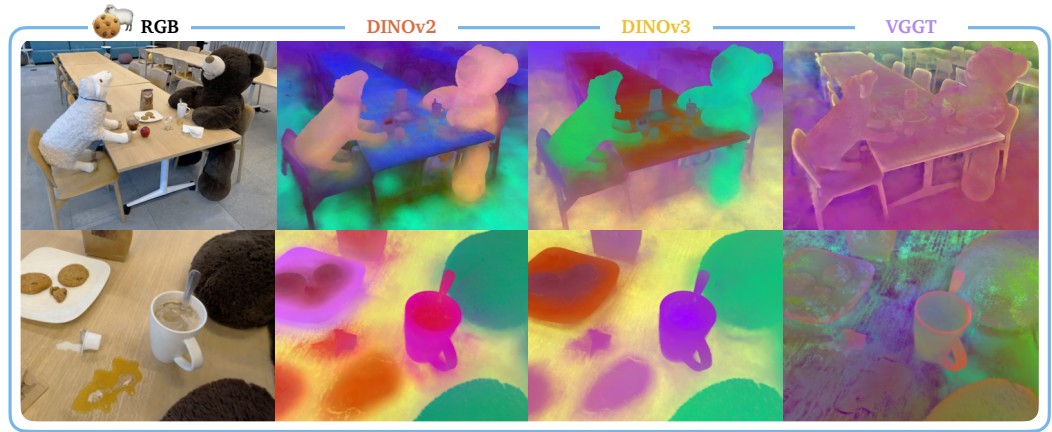

Figure 27: Semantic content of distilled features visualized for NERF for the LERF teatime scene.

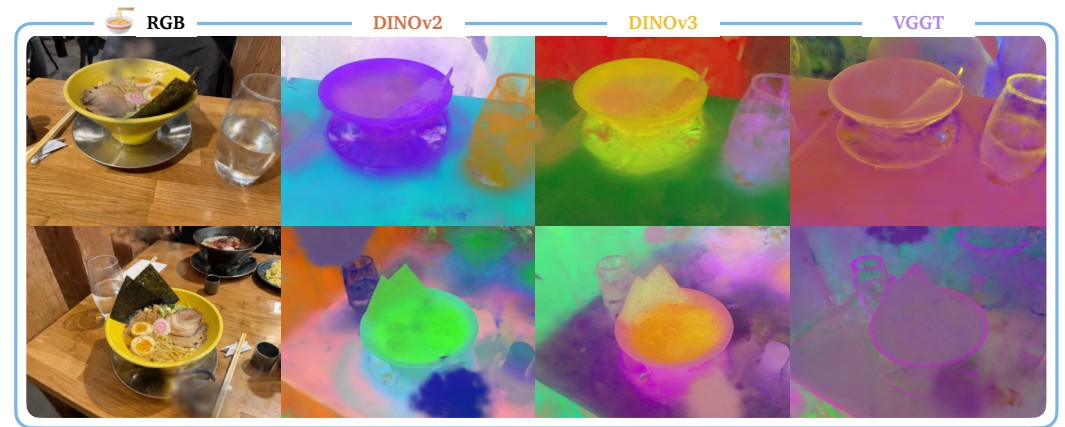

Figure 28: Semantic content of distilled features visualized for NERF for the LERF ramen scene.

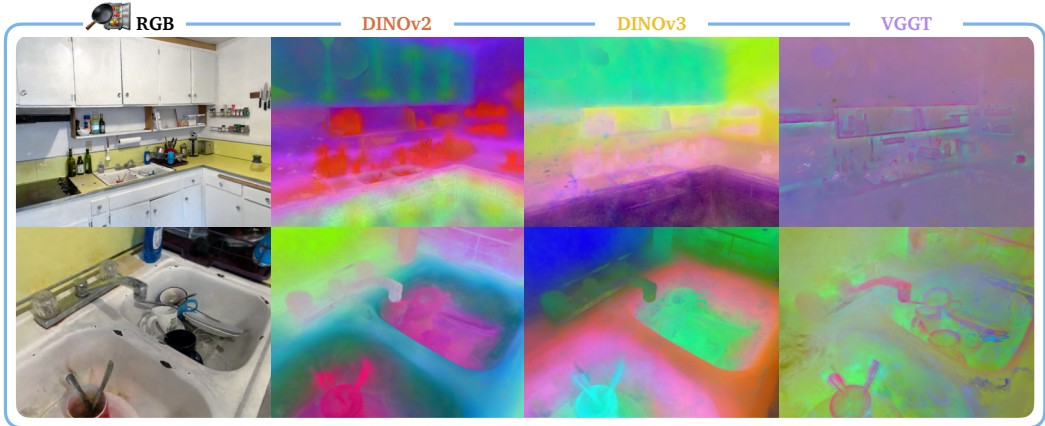

Figure 29: Semantic content of distilled features visualized for NERF for the LERF kitchen scene.

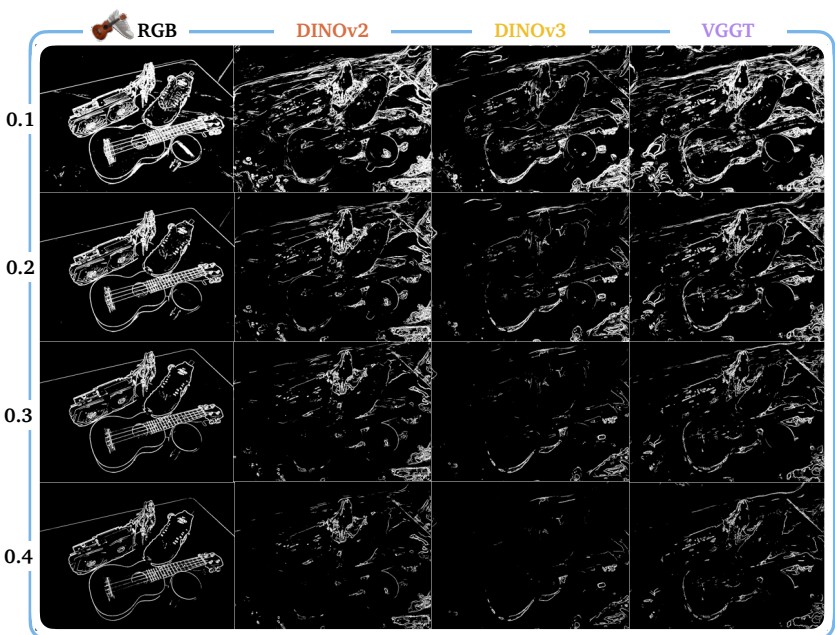

Figure 30: GFF of geometry-grounded and visual-semantic features visualized for GS for the 3D-OVS table scene.

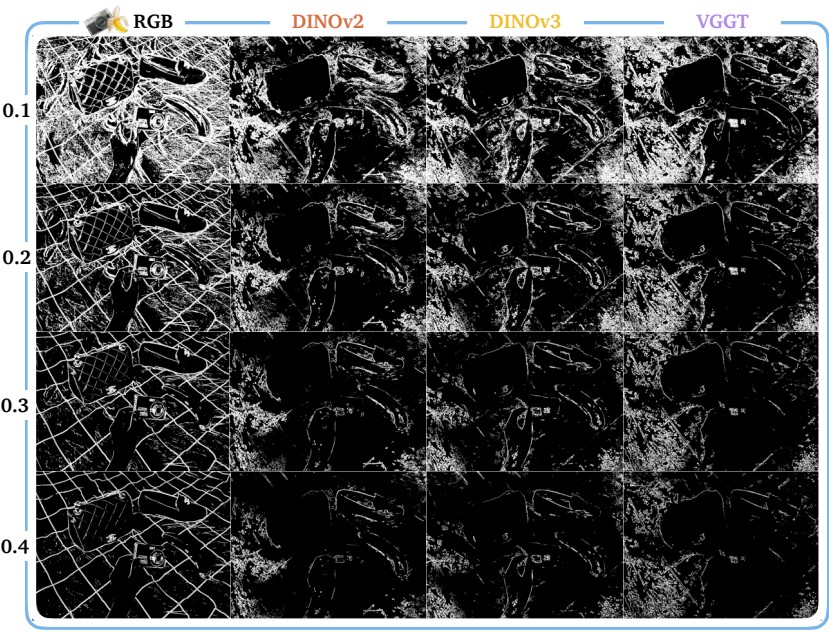

Figure 31: GFF of geometry-grounded and visual-semantic features visualized for GS for the 3D-OVS bed scene.

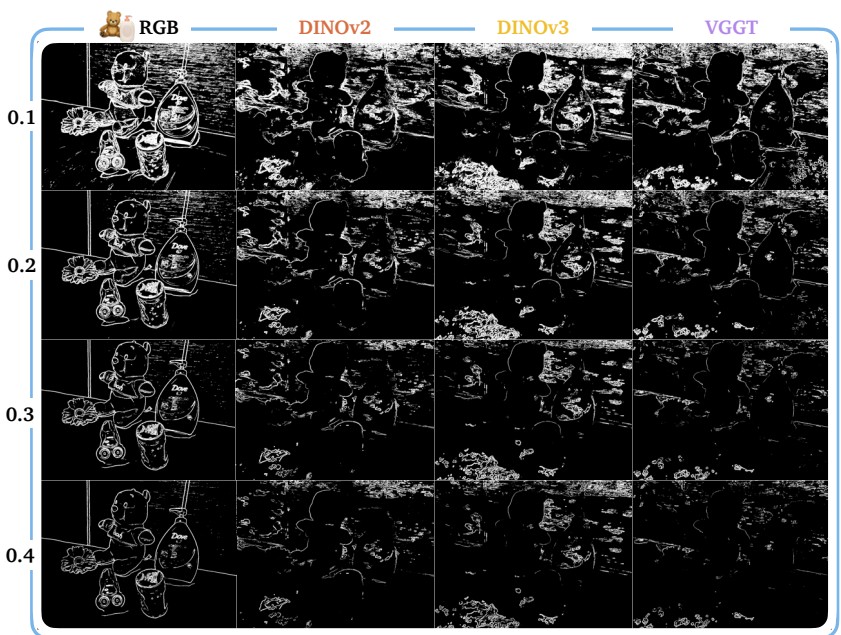

Figure 32: GFF of geometry-grounded and visual-semantic features visualized for GS for the 3D-OVS desk scene.

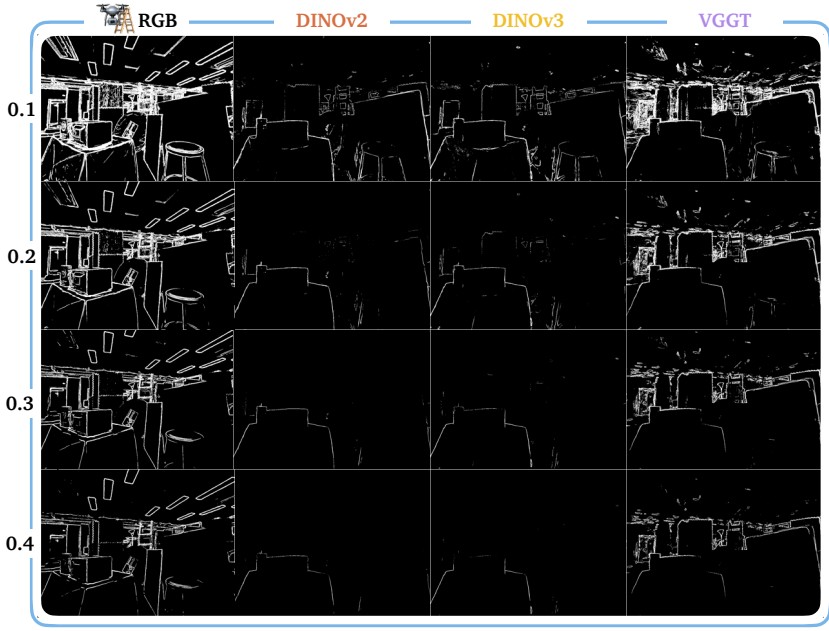

Figure 33: GFF of geometry-grounded and visual-semantic features visualized for GS for the drone scene.

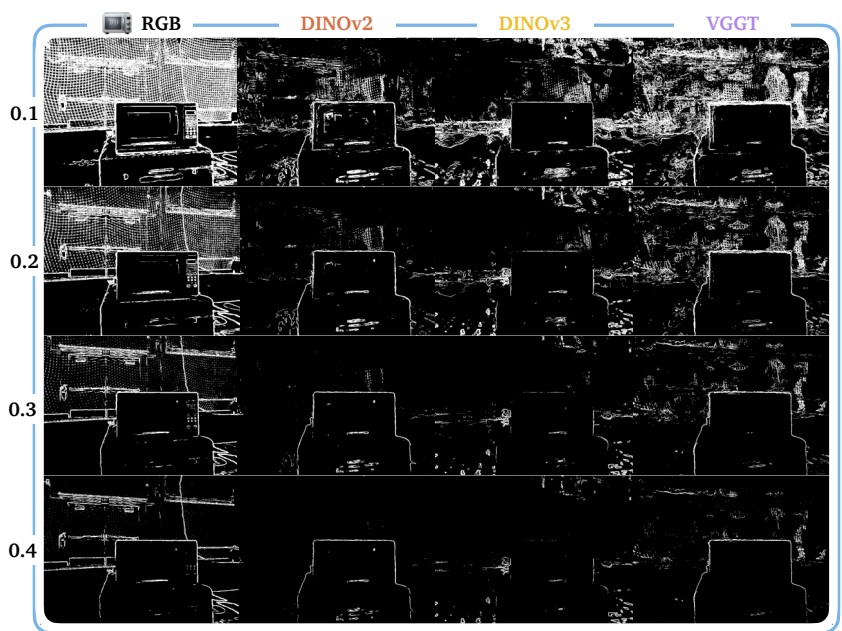

Figure 34: GFF of geometry-grounded and visual-semantic features visualized for GS for the quadruped kitchen scene.

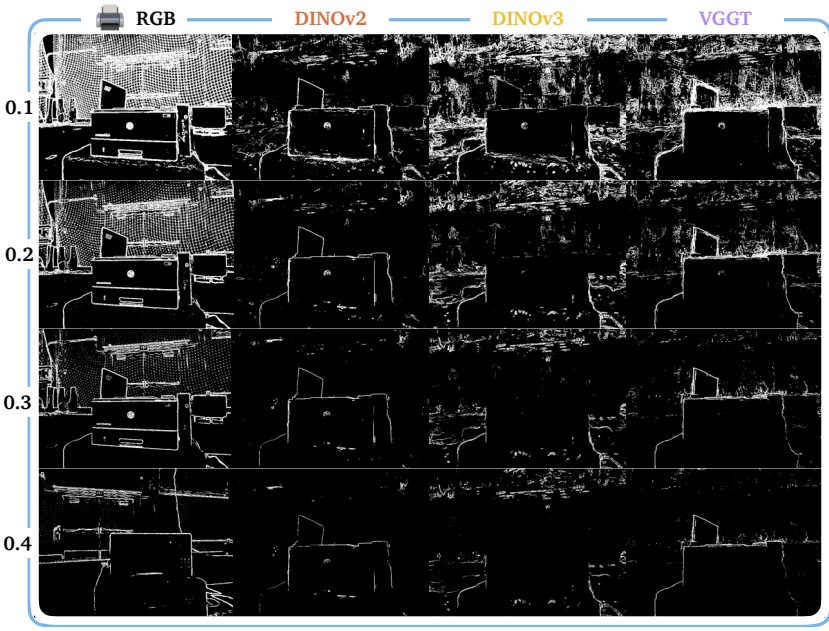

Figure 35: GFF of geometry-grounded and visual-semantic features visualized for GS for the quadruped office scene.

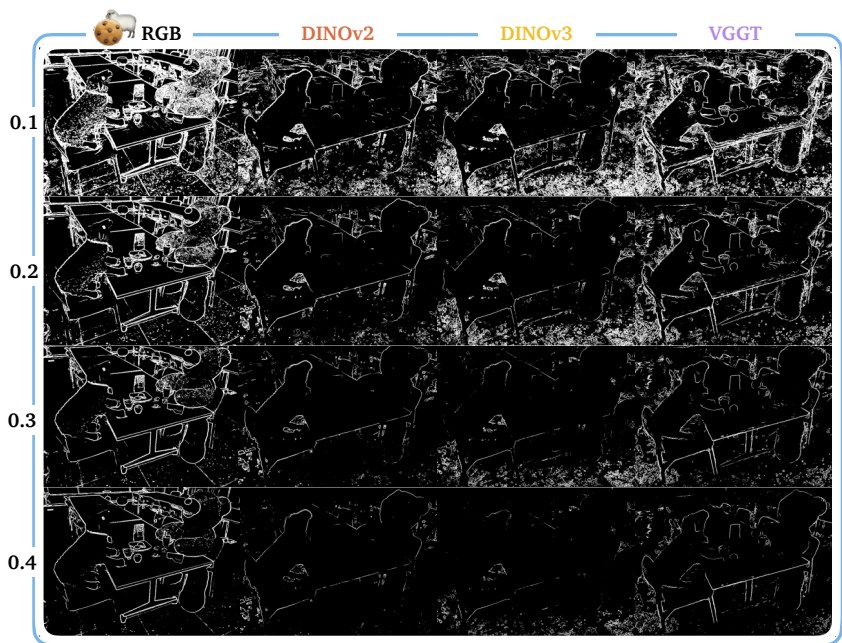

Figure 36: GFF of geometry-grounded and visual-semantic features visualized for GS for the LERF teatime scene.

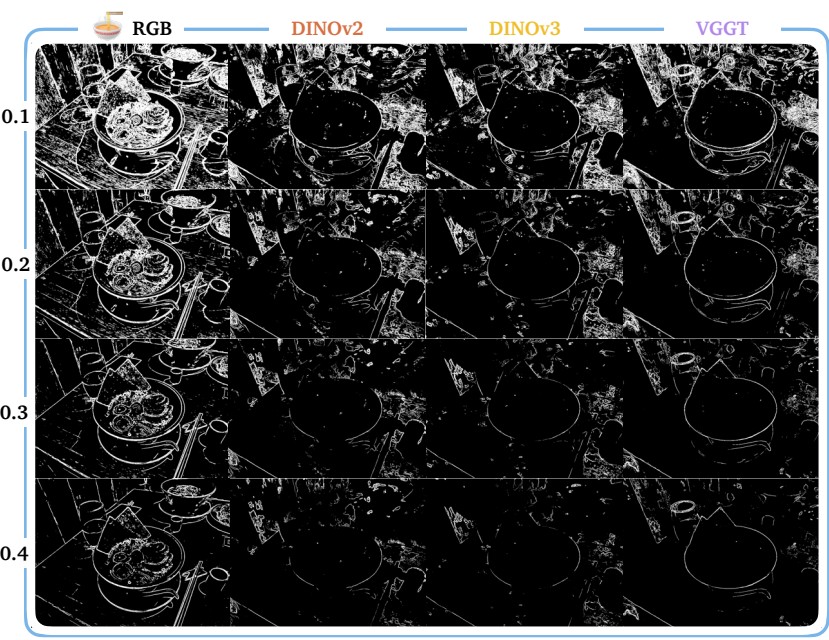

Figure 37: GFF of geometry-grounded and visual-semantic features visualized for GS for the LERF ramen scene.

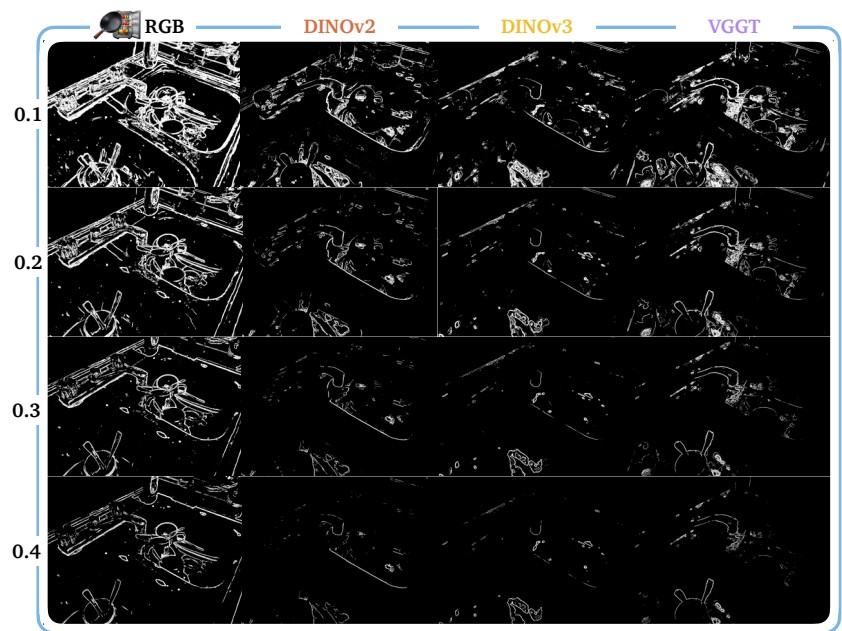

Figure 38: GFF of geometry-grounded and visual-semantic features visualized for GS for the LERF kitchen scene.

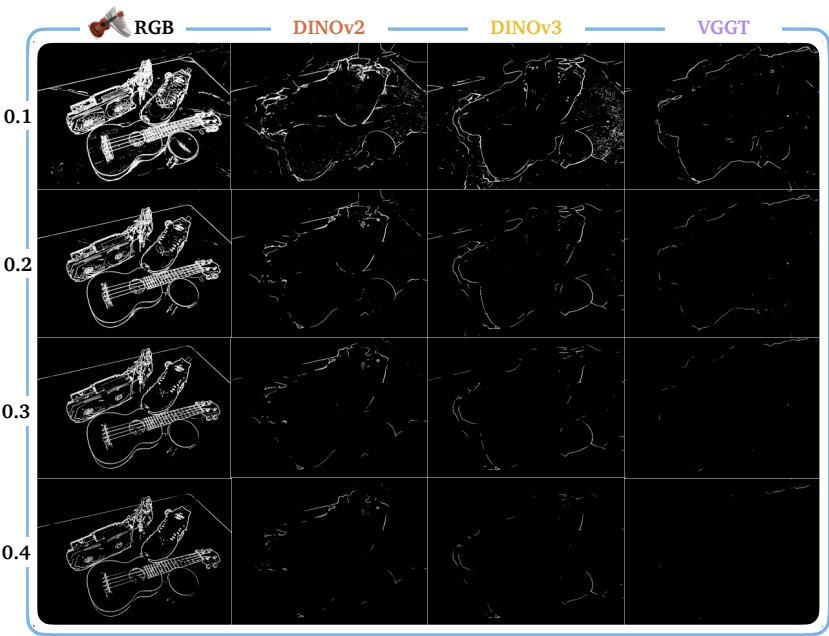

Figure 39: GFF of geometry-grounded and visual-semantic features visualized for NERF for the 3D-OVS table scene.

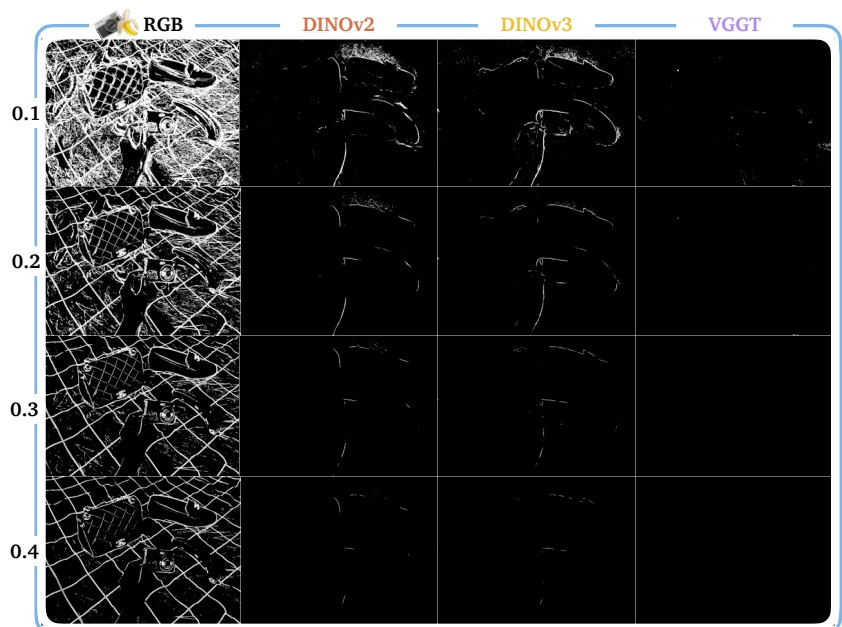

Figure 40: GFF of geometry-grounded and visual-semantic features visualized for NERF for the 3D-OVS bed scene.

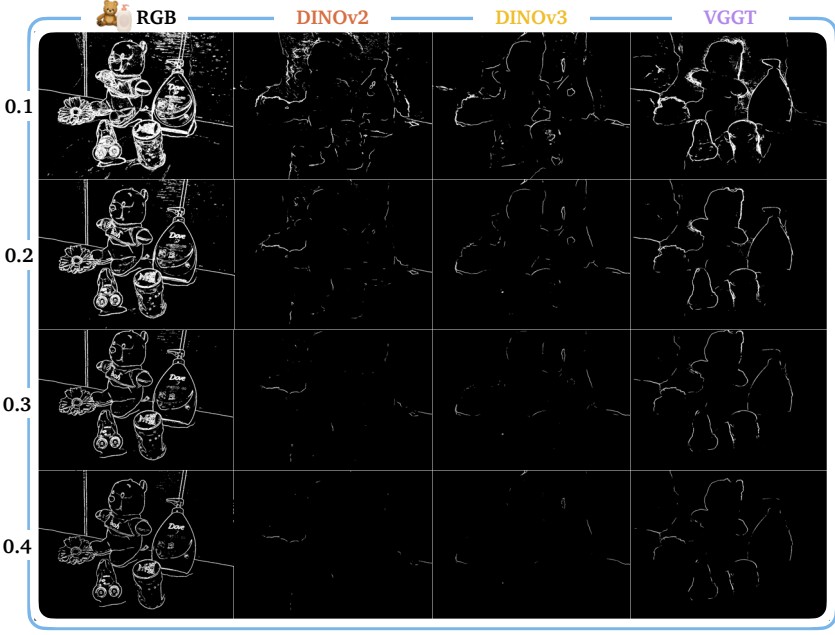

Figure 41: GFF of geometry-grounded and visual-semantic features visualized for NERF for the 3D-OVS desk scene.

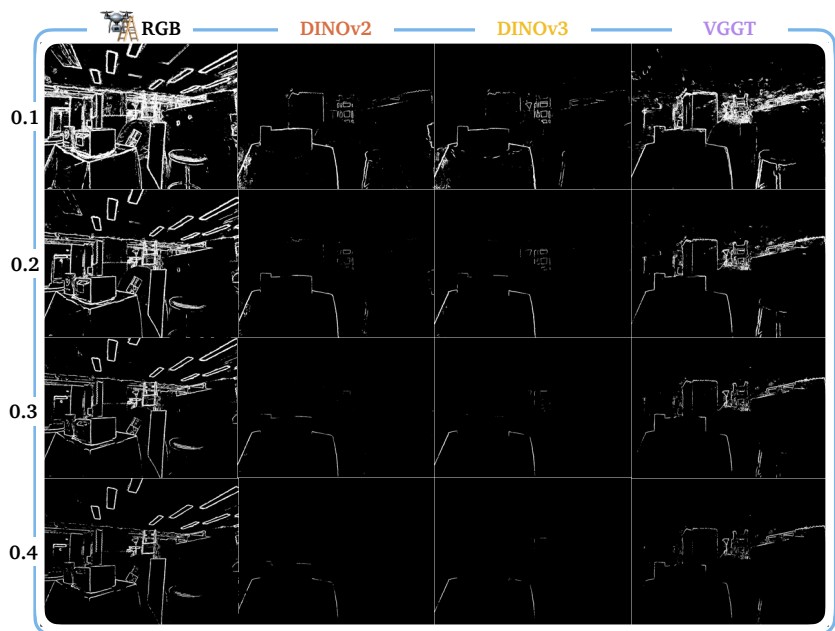

Figure 42: GFF of geometry-grounded and visual-semantic features visualized for NERF for the drone scene.

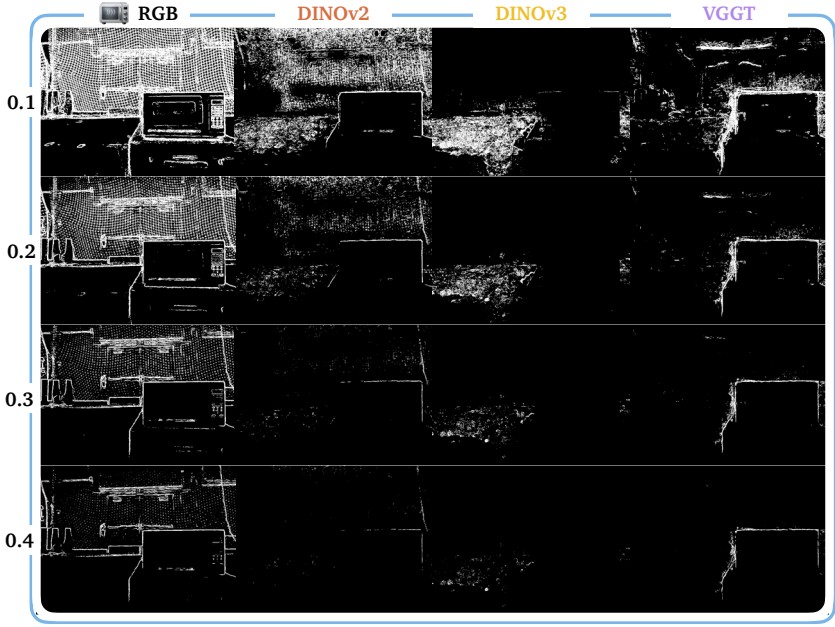

Figure 43: GFF of geometry-grounded and visual-semantic features visualized for NERF for the quadruped kitchen scene.

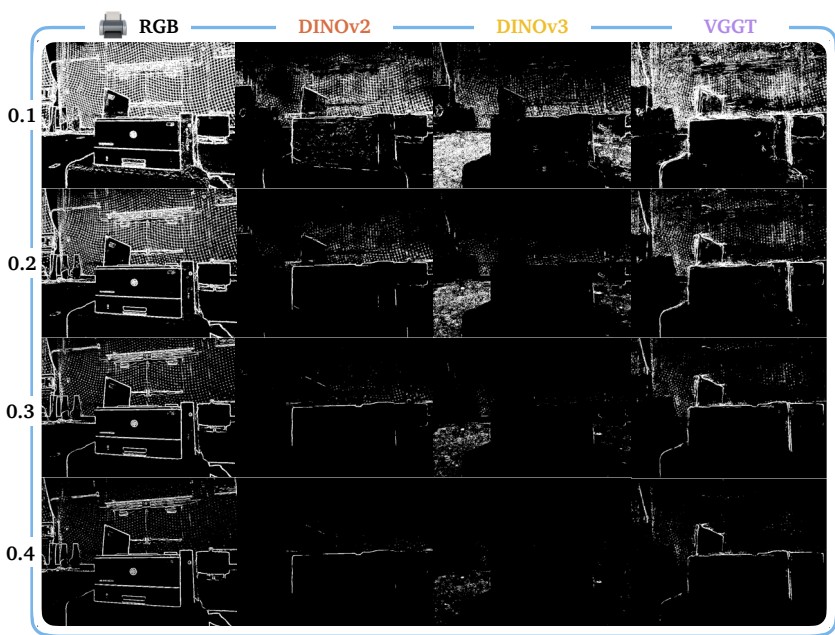

Figure 44: GFF of geometry-grounded and visual-semantic features visualized for NERF for the quadruped office scene.

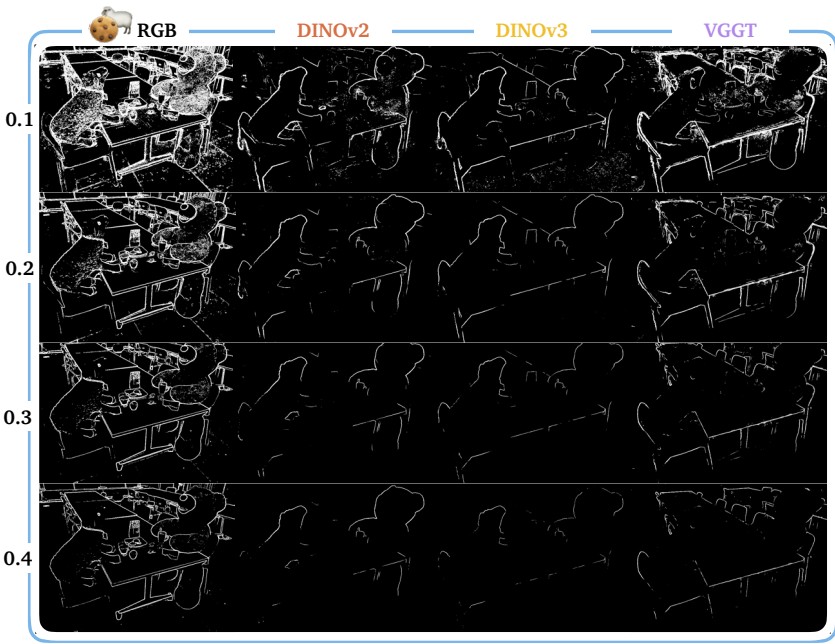

Figure 45: GFF of geometry-grounded and visual-semantic features visualized for NERF for the LERF teatime scene.

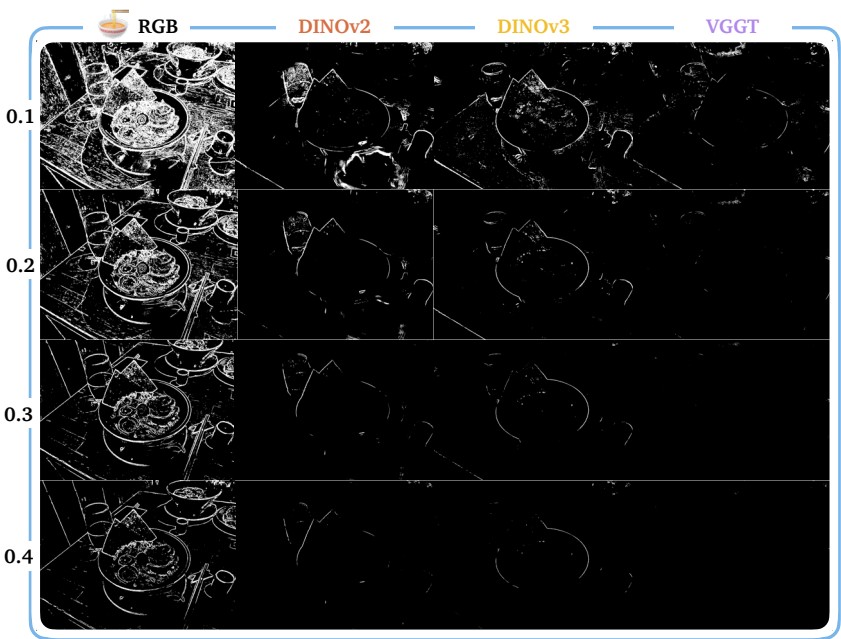

Figure 46: GFF of geometry-grounded and visual-semantic features visualized for NERF for the LERF ramen scene.

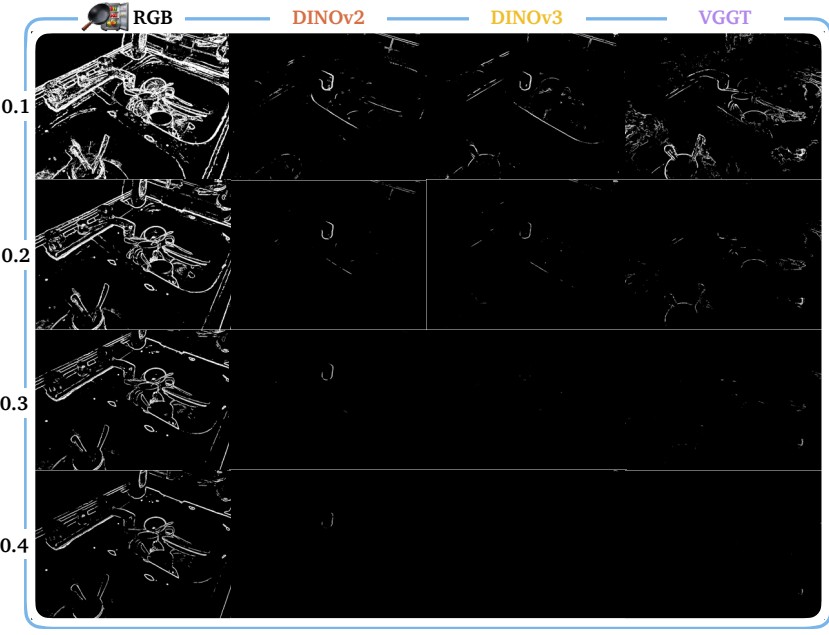

Figure 47: GFF of geometry-grounded and visual-semantic features visualized for NERF for the LERF kitchen scene.

## A.5    ADDITIONAL SEMANTIC LOCALIZATION RESULTS

We present the quantitative metrics for each scene for DINOv2, DINOv3, and VGGT in Figure 48, highlighting that there's no notable difference in performance among these encoders for the semantic localization task.

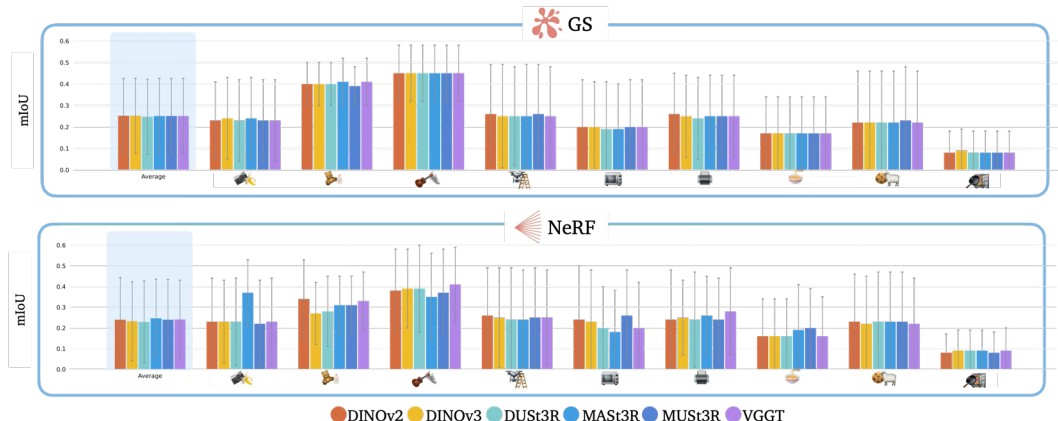

Figure 49: Quantitative result for the mIoU metric for semantic localization across each scene.

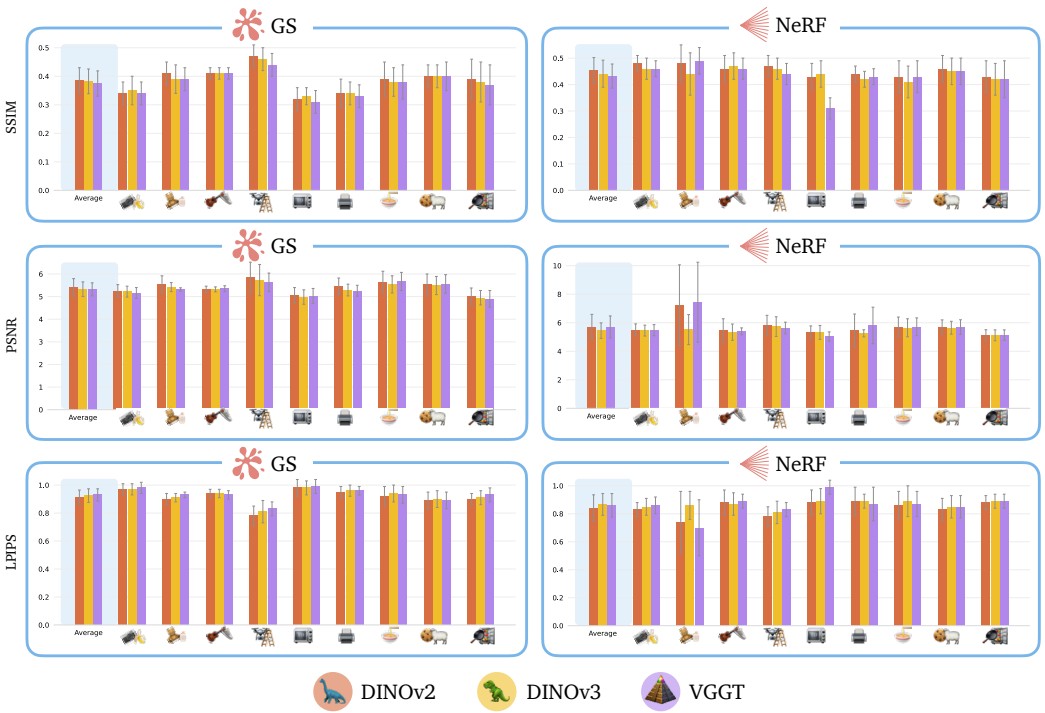

Figure 48: Quantitative results for the three metrics (SSIM, PSNR, LPIPS) of semantic object localization across each scene.

We provide the mIoU metric for all methods in Figure 49, where the threshold for language embedding similarity is set to 0.9. Similarly, we do not see notable performance difference among the vision encoders on average. We found the mIoU to be less informative in our settings, particularly in terms of the resolution achieved, given that notable similarity in the segmentation outputs of the different methods and the requirement of a threshold to define mask. Given that we use only CLIP for segmentation guidance for simplicity, we observe relatively lower mIoU scores.

Below, we include semantic localization visualizations, sampling one or two camera views from each scene.

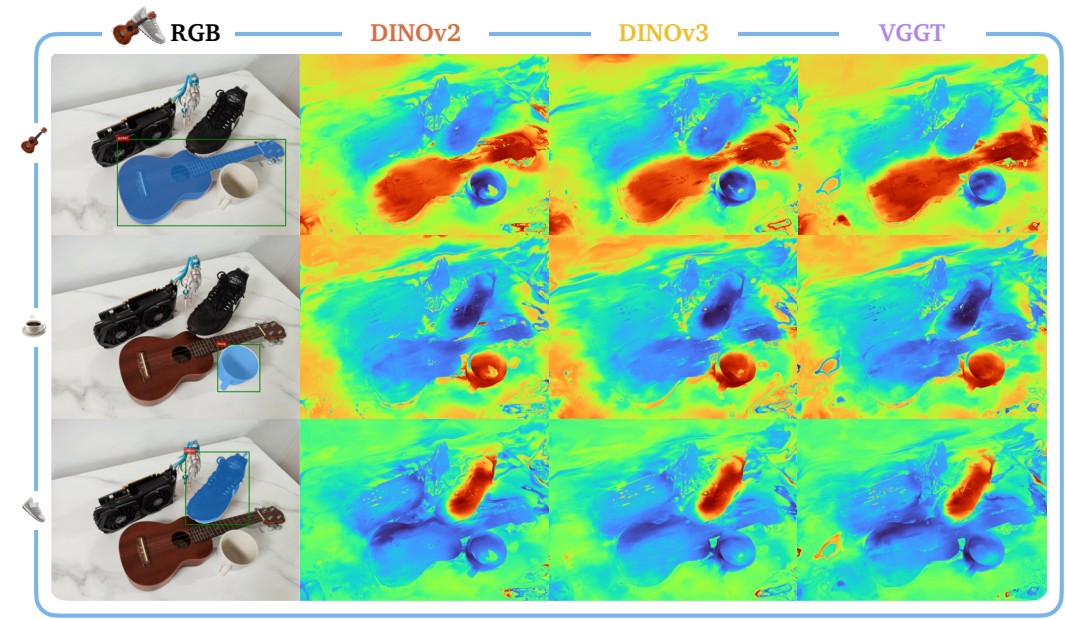

Figure 50: Semantic object localization visualized for GS for the 3D-OVS table scene.

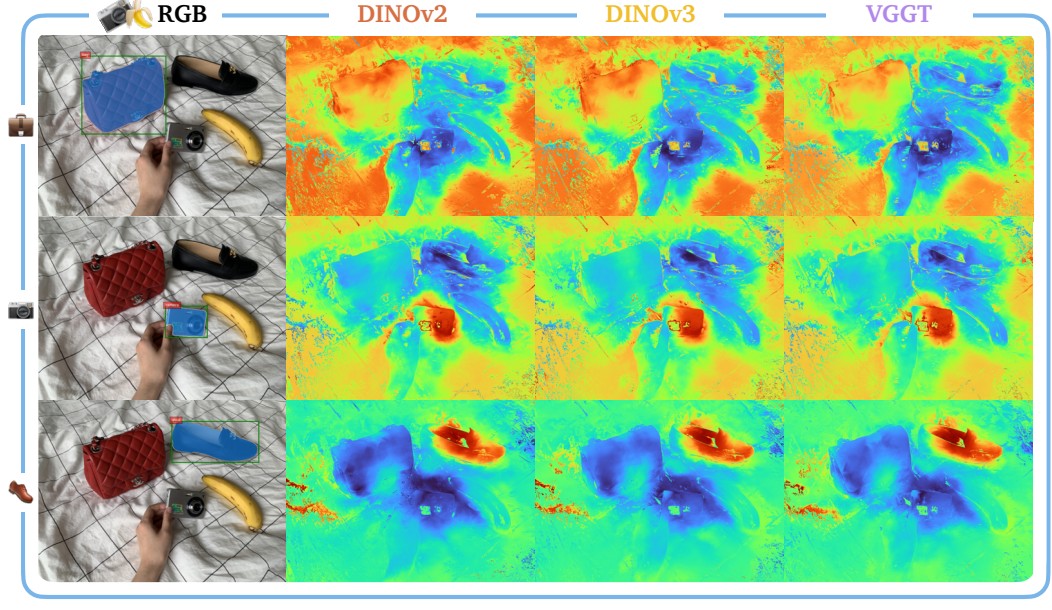

Figure 51: Semantic object localization visualized for GS for the 3D-OVS bed scene.

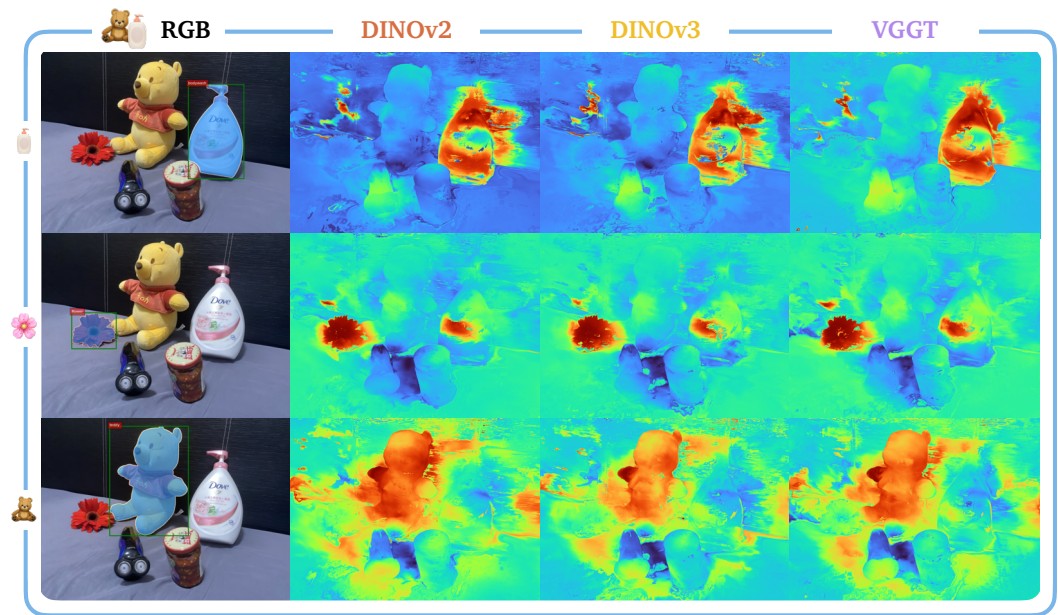

Figure 52: Semantic object localization visualized for GS for the 3D-OVS covered desk scene.

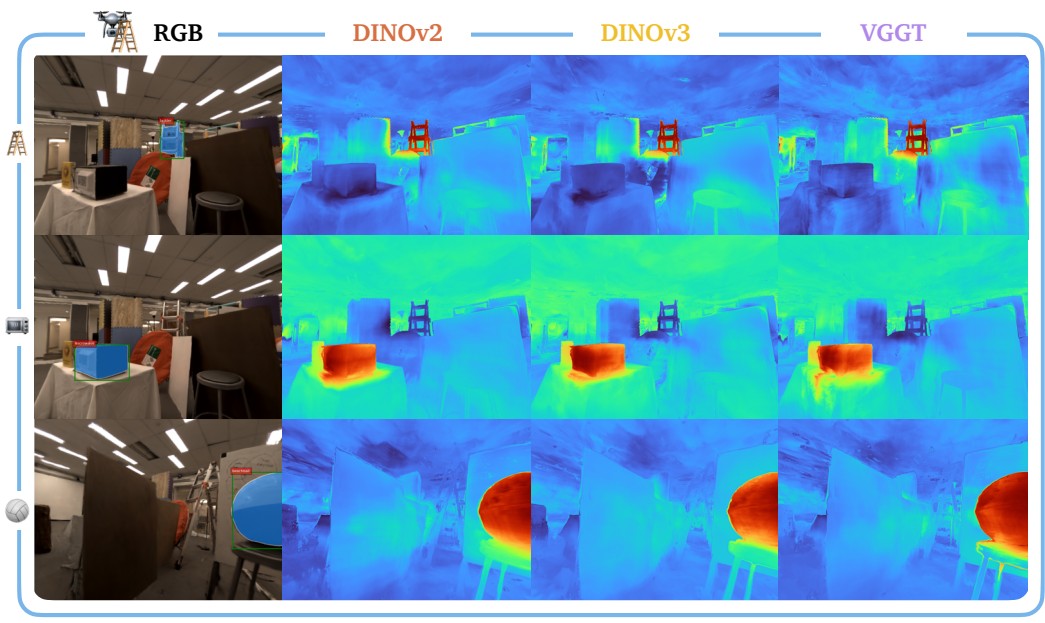

Figure 53: Semantic object localization visualized for GS for the drone scene.

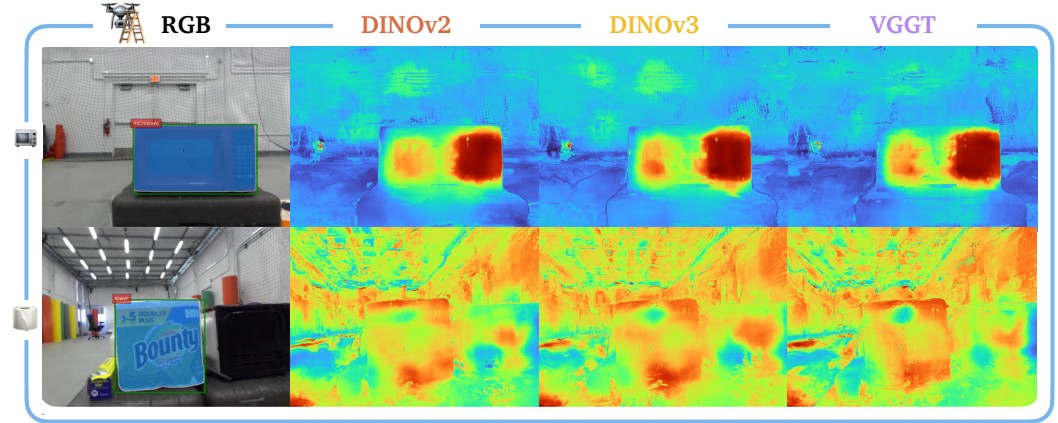

Figure 54: Semantic object localization visualized for GS for the quadruped kitchen scene.

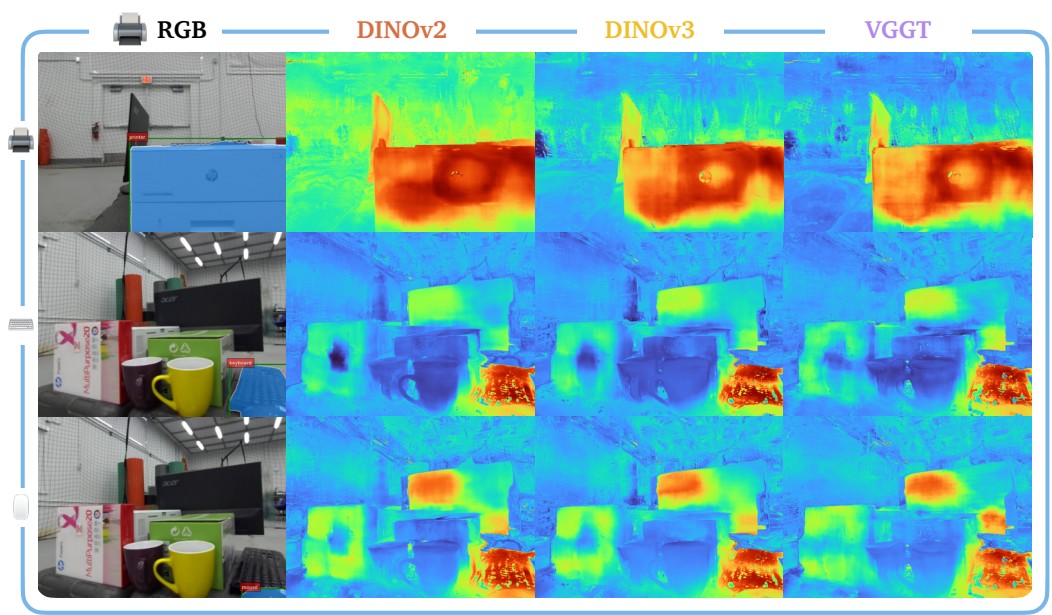

Figure 55: Semantic object localization visualized for GS for the 3D-OVS bed scene.

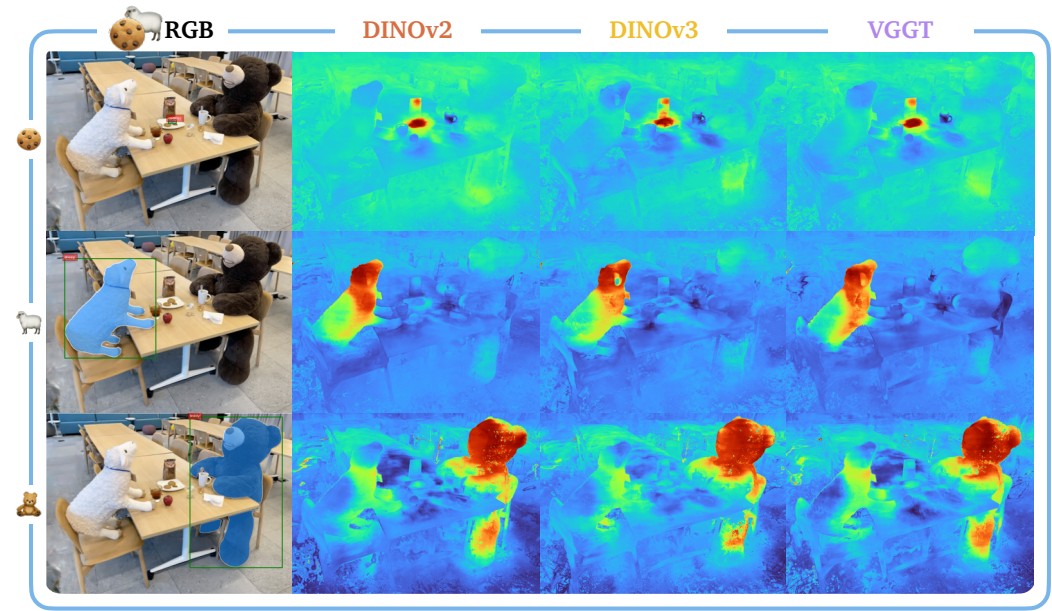

Figure 56: Semantic object localization visualized for GS for the LERF teatime scene.

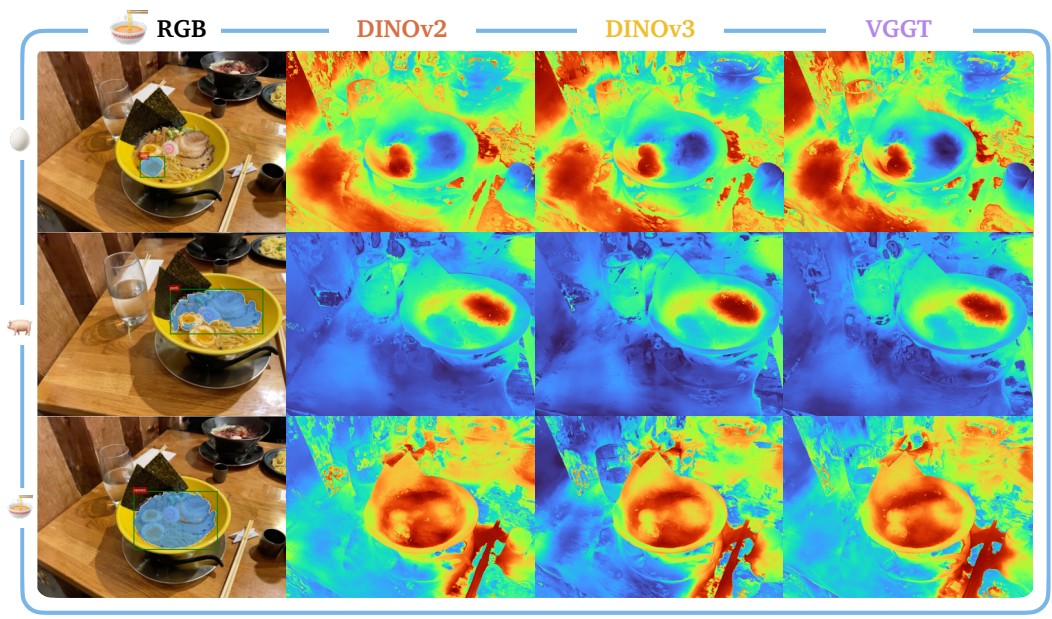

Figure 57: Semantic object localization visualized for GS for the LERF ramen scene.

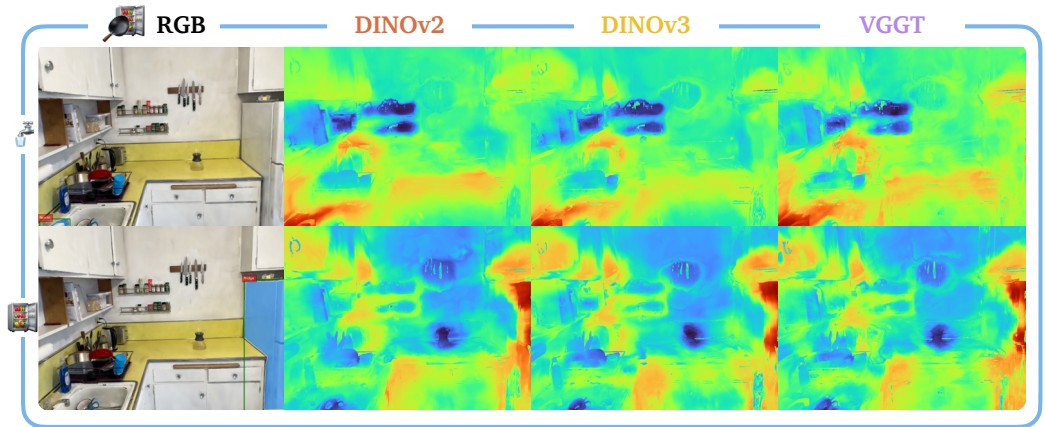

Figure 58: Semantic object localization visualized for GS for the LERF kitchen scene.

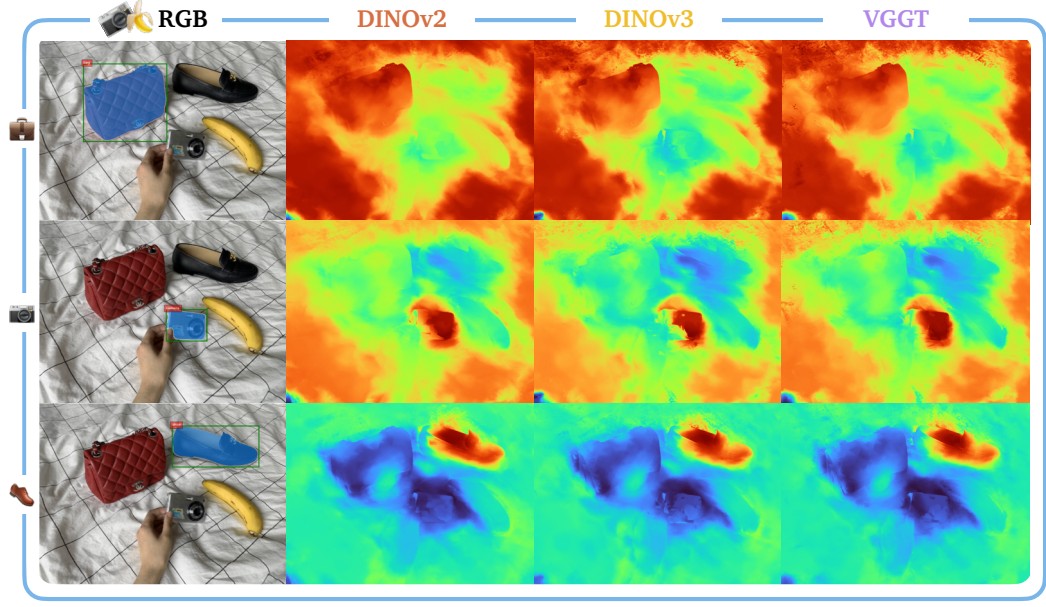

Figure 59: Semantic object localization visualized for NERF for the 3D-OVS table scene.

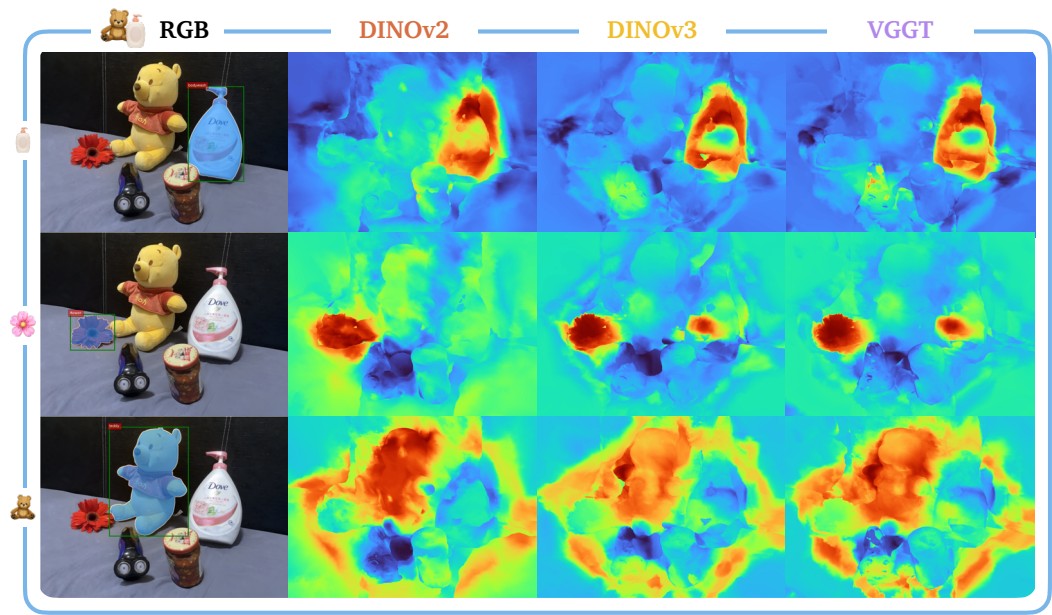

Figure 60: Semantic object localization visualized for NERF for the 3D-OVS bed scene.

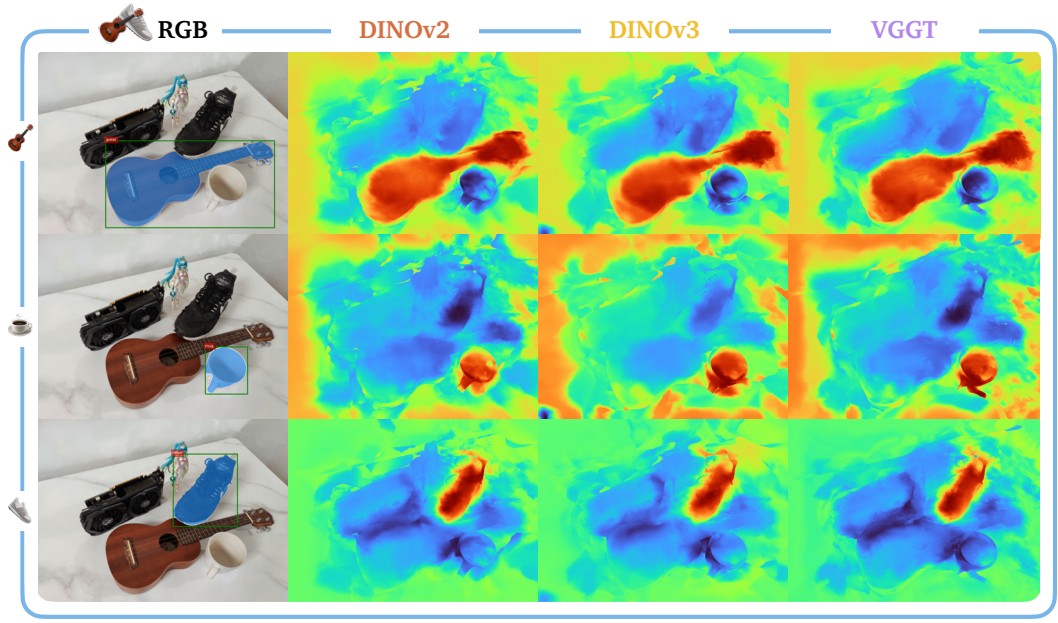

Figure 61: Semantic object localization visualized for NERF for the 3D-OVS covered desk scene.

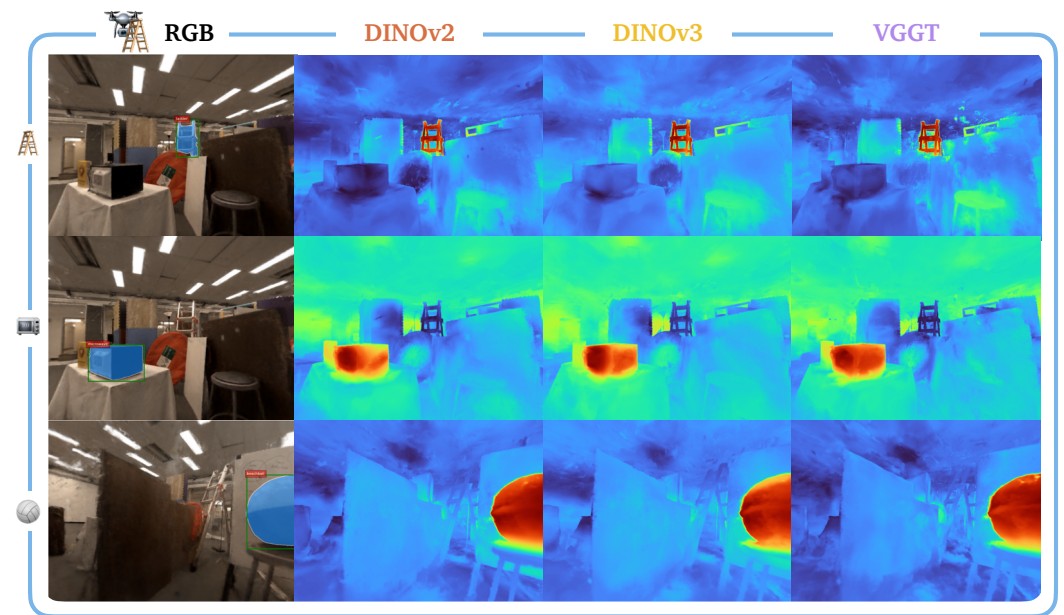

Figure 62: Semantic object localization visualized for NERF for the drone scene.

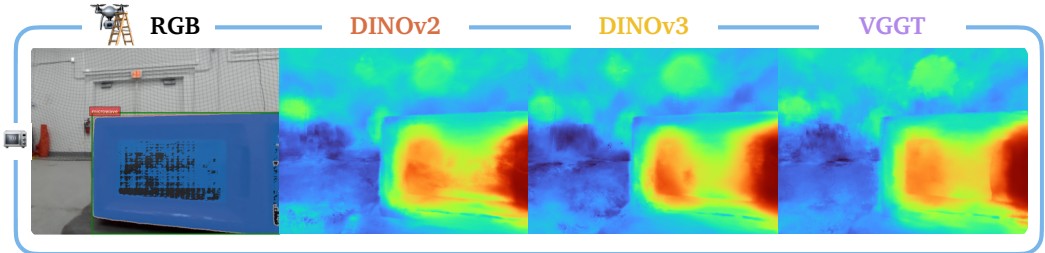

Figure 63: Semantic object localization visualized for NERF for the quadruped kitchen scene.

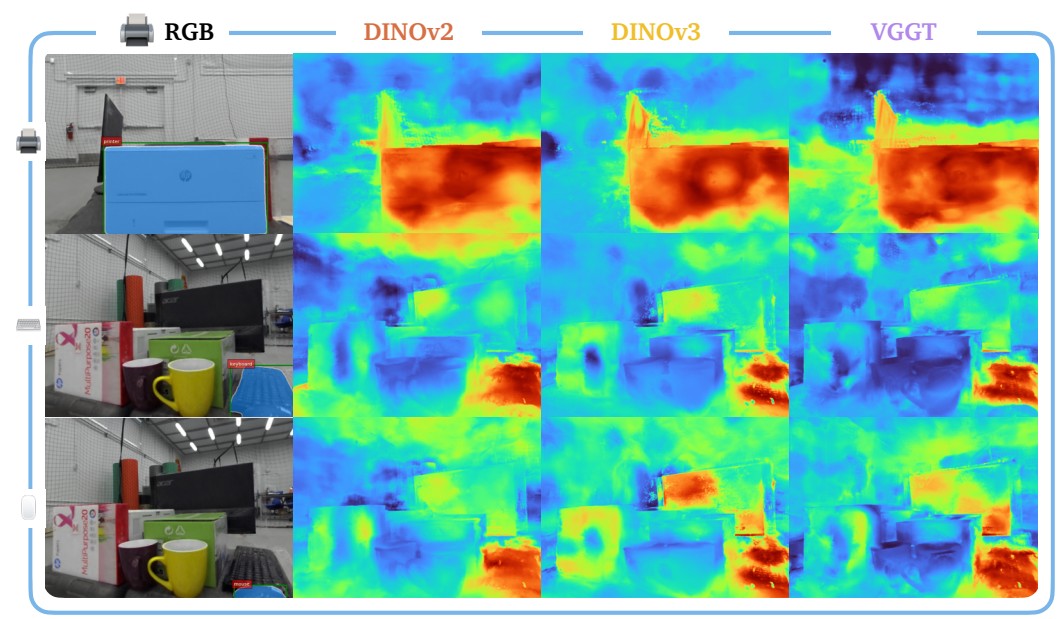

Figure 64: Semantic object localization visualized for NERF for the 3D-OVS bed scene.

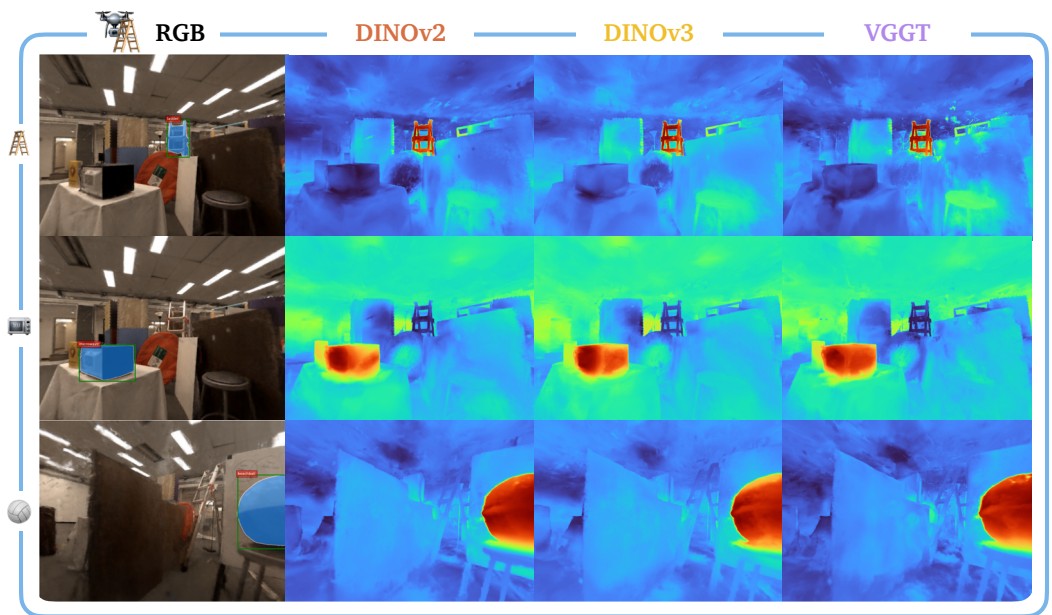

Figure 65: Semantic object localization visualized for NERF for the LERF teatime scene.

## A.6 ADDITIONAL RADIANCE FIELD INVERSION RESULTS

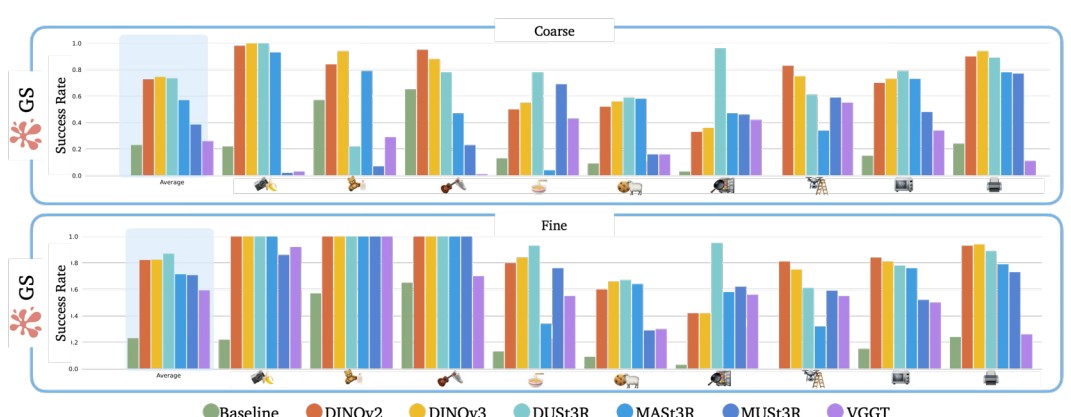

Figure 66: (GS) RF Inversion success rate

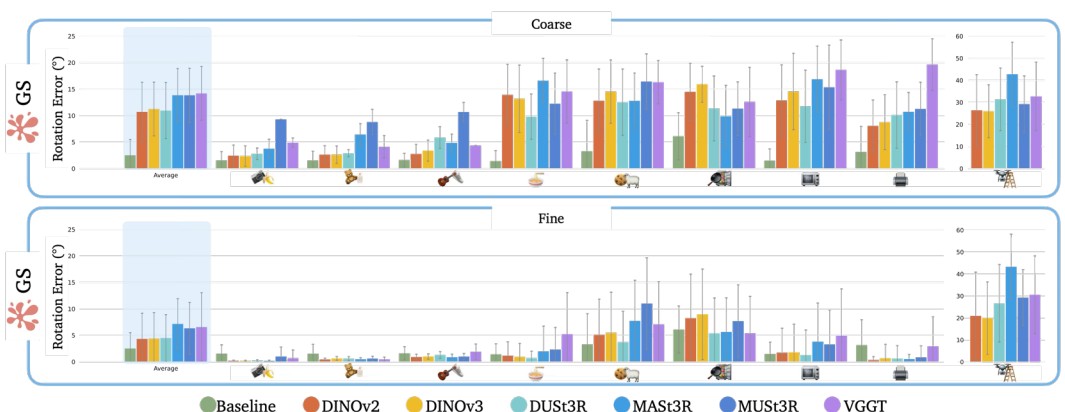

Figure 67: (GS) RF Inversion rotation error

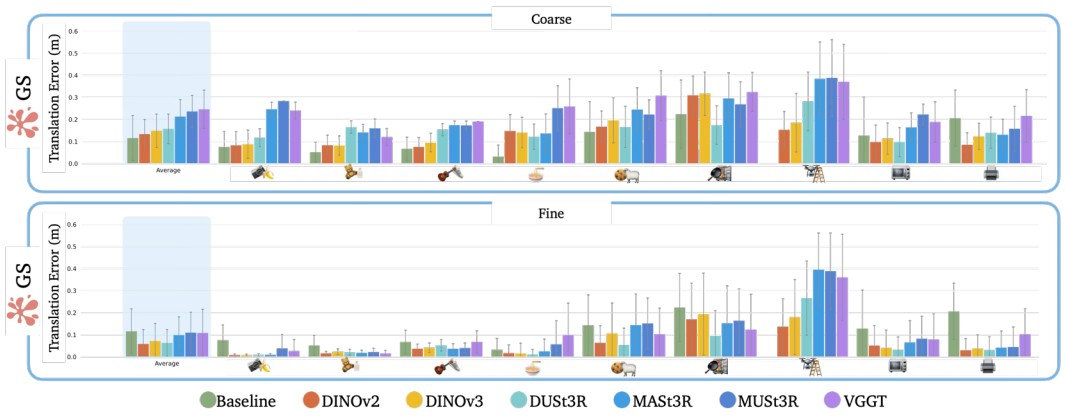

Figure 68: (GS) RF Inversion translation error

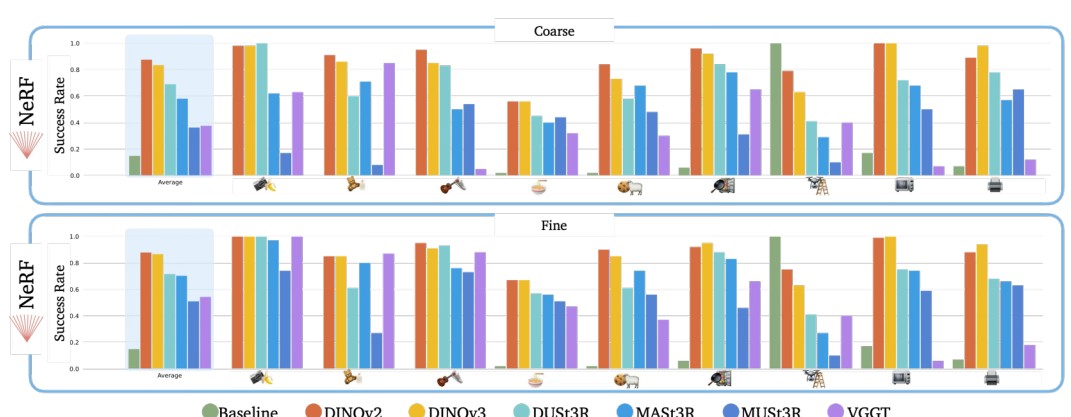

Figure 69: (NeRF) RF Inversion success rate

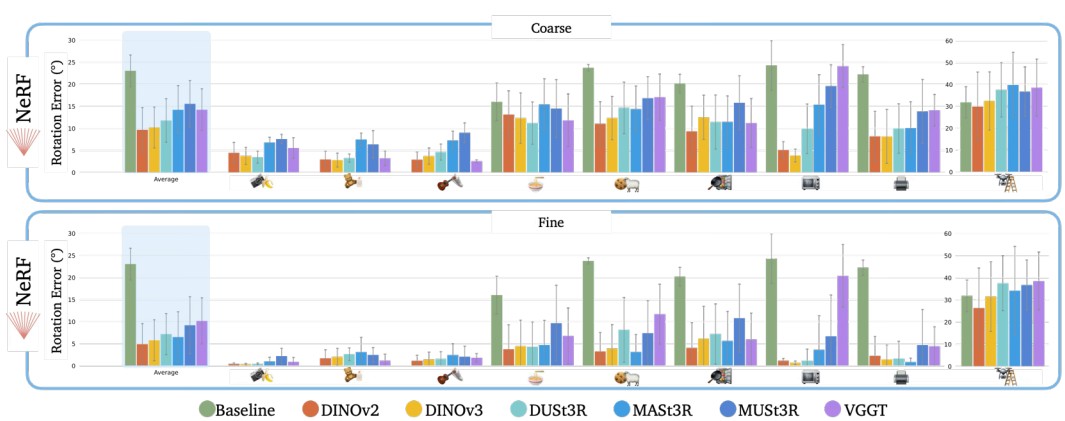

Figure 70: (NeRF) RF Inversion rotation error

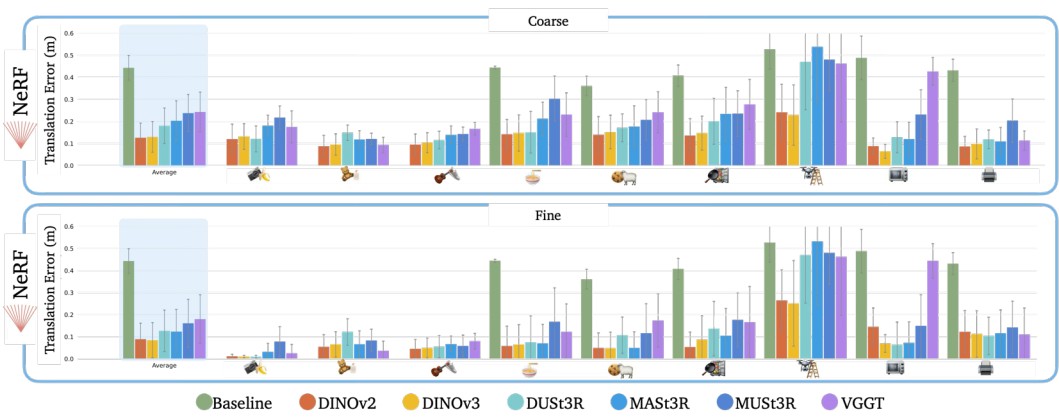

Figure 71: (NeRF) RF Inversion translation error