# OpenReview forum: "Geometry Meets Vision: Revisiting Pretrained Semantics in Distilled Fields"
_ICLR.cc/2026/Conference — Submitted to ICLR 2026_

### Official Review · Reviewer_x4tX · 2025-10-18

**Soundness:** 3
**Presentation:** 3
**Contribution:** 3
**Rating:** 4
**Confidence:** 4

**Summary:**

This paper conducts an empirical study to compare the effectiveness of visual-only semantic features (from DINOv2, DINOv3) against visual-geometry semantic features (from VGGT) when distilled into 3D radiance fields (Gaussian Splatting and NeRFs). The investigation is structured around three core questions relevant to robotics applications:
1. Do visual-geometry features contain higher-fidelity spatial content?
2. Does geometry-grounding improve semantic object localization?
3. Can visual-geometry features enable higher-accuracy radiance field inversion (i.e., camera pose estimation)?

To facilitate the third question, the authors propose SPINE, a novel framework for radiance field inversion that does not require an initial pose estimate. The key findings are that while geometry-grounded features (VGGT) do capture finer geometric details, they do not offer an advantage in semantic localization and surprisingly underperform visual-only features (DINOv2) in the task of pose estimation. The authors conclude that visual-only features currently offer greater versatility for downstream tasks in distilled fields.

**Strengths:**

- The paper addresses a fundamental and highly relevant question: what kind of pretrained features are most effective for 3D scene understanding in the context of radiance fields? Comparing modern visual-only foundation models like DINO with emerging visual-geometry models like VGGT provides valuable insights for the community.
- The study is well-organized, comparing two types of features on two different radiance field representations (NeRF and GS) across three distinct and important downstream tasks.
- Beyond a comparative study, the paper introduces SPINE, a novel method for radiance field inversion that works without an initial camera pose guess.
- The results are somewhat counter-intuitive; one might expect geometry-grounded features to excel at a geometric task like pose estimation. The finding that visual-only features from DINOv2 perform better is surprising and important.

**Weaknesses:**

- **Overly Broad Claims** from a Single Model: The paper's claims about "visual-geometry features" as a general class are based on experiments with a single model, VGGT. While VGGT is a strong representative, it is possible that its specific pre-training objective is what limits its versatility, rather than the principle of geometry-grounding itself. The conclusions should be carefully worded to reflect that the findings are specific to the models tested, avoiding generalization to all possible geometry-grounding techniques.
- The paper presents SPINE as a method for pose estimation, but its training protocol and generalization capabilities are not clearly described.
- The authors use perceptual metrics (SSIM, PSNR, LPIPS) to evaluate semantic localization accuracy. While the rationale is noted, this is a departure from standard practice for localization/segmentation tasks. These metrics measure the similarity of the relevancy heatmaps, not how accurately the target object is isolated. Reporting results across a range of thresholds with mIoU would make the results more convincing.

**Questions:**

The paper tackles a very interesting problem with a well-structured set of experiments and introduces a novel pose estimation framework (SPINE). The findings are surprising and thought-provoking. However, the work is held back by a few significant weaknesses: the conclusions about geometry-grounded features are drawn from a **single model**, and the evaluation for semantic localization uses non-standard metrics in my opinion. These issues prevent a confident recommendation for acceptance in its current form. I believe the paper has high potential, and I would be willing to reconsider my rating if the authors can address these points in a revision.

---

> ### Author Response · Authors · 2025-11-26
> **Response to Reviewer x4tX (Part I)**
>
> Thank you for recognizing the importance of our work. We are
> grateful for your valuable feedback, which we found to be really
> helpful and constructive. We group our response into the following
> sections:
>
> **Broad Claims.** We have included additional experiments with
> DUSt3R, MASt3R, and MUSt3R to broaden the scope of spatially
> grounded models. In Section 4, we provide a detailed discussion of
> the semantic content of these models. We find that our conclusions
> on VGGT generalize to these models. For example, these model
> attend to structural information in their input images, discarding
> object-level semantic information for more fine-grained geometric
> details, which could hurt their performance in tasks that more heavily
> depend on object-level semantic information.
>
> **SPINE.** We clarify that radiance field inversion is a well-established
> research area [10 ]. Our work follows standard practice in this research
> area, evaluating SPINE across many benchmark scenes. We have
> revised our discussion of radiance field inversion to better discuss the
> nuances of this task compared to other related tasks. We emphasize
> that although radiance field inversion is related to a number of
> other problems, e.g., camera relocalization [ 5 , 6, 7] and 3D scene
> reconstruction [8 , 9], a few key characteristics distinguish radiance
> field inversion from these tasks. Specifically, camera relocalization
> methods (e.g., [ 5]) generally require lots of training data (on the
> order of thousands to tens of thousands). In contrast, radiance field
> inversion typically utilizes training data with fewer than a hundred (or
> a few hundred) samples, a scale impractical for camera relocalization
> methods. Likewise, radiance field inversion methods computes poses
> in a global reference frame, unlike 3D scene reconstruction methods.
> Like all camera relocalization and radiance field inversion methods,
> SPINE is trained per scene. However, SPINE is still applicable to
> real robotic deployments, as demonstrated in prior work [10].
>
> We would also like to clarify that SPINE is a secondary
> contribution in our work. Nonetheless, SPINE turns out to be a
> significant contribution that addresses one of the major drawbacks
> of existing radiance field inversion methods, by eliminating the need
> for an initial guess. If this solution was obvious, existing methods
> would have implemented it to address their weaknesses. However,
> no prior work has developed a solution to eliminate reliance on an
> initial guess, underscoring the importance of our contribution.
>
> **Perceptual Metrics.** We selected these perceptual metrics as a
> native solution to accurately measuring the semantic localization
> accuracy without introducing additional hyperparameters that could
> influence the reliability of the results. We found the mIoU to be less
> informative in our settings, particularly in terms of the resolution
> achieved, given that notable similarity in the segmentation outputs
> of the different methods. Since the methods only rely on CLIP for
> segmentation guidance, we obtain relatively lower mIoU scores. For
> completeness, we have provided the mIoU plots in Appendix A.5.
>
> Thank you for your feedback. We hope our revisions and response
> has addressed your concerns about the paper. We would be glad if
> you could consider updating your score to reflect the revision.

---

> > ### Author Response · Authors · 2025-11-26
> > **Response to Reviewer x4tX (Part II)**
> >
> > [1] Cheng Chi, Zhenjia Xu, Siyuan Feng, Eric Cousineau, Yilun Du, Benjamin Burchfiel, Russ Tedrake, and Shuran Song. Diffusion policy: Visuomotor policy learning via action diffusion. The International Journal of Robotics Research, 44(10-11):1684–1704, 2025.
> >
> > [2] Siddharth Karamcheti, Suraj Nair, Ashwin Balakrishna, Percy Liang, Thomas Kollar, and Dorsa Sadigh. Prismatic VLMs: Investigating the Design Space of Visually-Conditioned Language Models. In International Conference on Machine Learning (ICML), volume 235, pages 11197–11210. PMLR, 2024.
> >
> > [3] Huang Huang, Fangchen Liu, Letian Fu, Tingfan Wu, Mustafa Mukadam, Jitendra Malik, Ken Goldberg, and Pieter Abbeel. Otter: A vision-language-action model with text-aware visual feature extraction. arXiv preprint arXiv:2503.03734, 2025.
> >
> > [4] Moo Jin Kim, Karl Pertsch, Siddharth Karamcheti, Ted Xiao, Ashwin Balakrishna, Suraj Nair, Rafael Rafailov, Ethan P Foster, Pannag R Sanketi, Quan Vuong, et al. Openvla: An open-source vision-language-action model. In Conference on Robot Learning, pages 2679–2713. PMLR, 2025.
> >
> > [5] Alex Kendall, Matthew Grimes, and Roberto Cipolla. Posenet: A convolutional network for real-time 6-dof camera relocalization. In Proceedings of the IEEE international conference on computer vision, pages 2938–2946, 2015.
> >
> > [6] Fei Xue, Xin Wu, Shaojun Cai, and Junqiu Wang. Learning multi- view camera relocalization with graph neural networks. In 2020 IEEE/CVF Conference on Computer Vision and Pattern Recognition (CVPR), pages 11372–11381. IEEE, 2020.
> >
> > [7] Qunjie Zhou, Maxim Maximov, Or Litany, and Laura Leal-Taix ´e. The nerfect match: Exploring nerf features for visual localization. In European Conference on Computer Vision, pages 108–127. Springer, 2024.
> >
> > [8] Vincent Leroy, Yohann Cabon, and Jerome Revaud. Grounding image matching innbsp;3d withnbsp;mast3r. In Computer Vision – ECCV 2024: 18th European Conference, Milan, Italy, September 29–October 4, 2024, Proceedings, Part LXXII, page 71–91, Berlin, Heidelberg, 2024. Springer-Verlag. ISBN 978-3-031-73219-5. doi: 10.1007/978-3-031-73220-1 5. URL https://doi.org/10.1007/ 978-3-031-73220-1 5.
> >
> > [9] Jianyuan Wang, Minghao Chen, Nikita Karaev, Andrea Vedaldi, Christian Rupprecht, and David Novotny. Vggt: Visual geometry grounded transformer. In Proceedings of the Computer Vision and Pattern Recognition Conference, pages 5294–5306, 2025.
> >
> > [10] Timothy Chen, Ola Shorinwa, Joseph Bruno, Aiden Swann, Javier Yu, Weijia Zeng, Keiko Nagami, Philip Dames, and Mac Schwager. Splat-nav: Safe real-time robot navigation in gaussian splatting maps. IEEE Transactions on Robotics, 2025.
> >
> > [11] Justin Kerr, Chung Min Kim, Ken Goldberg, Angjoo Kanazawa, and Matthew Tancik. Lerf: Language embedded radiance fields. In Proceedings of the IEEE/CVF international conference on computer vision, pages 19729–19739, 2023.
> >
> > [12] William Shen, Ge Yang, Alan Yu, Jansen Wong, Leslie Pack Kaelbling, and Phillip Isola. Distilled feature fields enable few-shot language-guided manipulation. arXiv preprint arXiv:2308.07931, 2023.

---

> > ### Comment · Reviewer_x4tX · 2025-11-28
> > **Response to Authors**
> >
> > Thank you for your response. I would like to check if you have included the Appendix in the revised manuscript as I cannot find the Appendix A.5 in the updated document. I would be happy to raise my score after checking the Appendix A.5.

---

> > > ### Author Response · Authors · 2025-11-28
> > > **Response to Reviewer x4tX**
> > >
> > > Thank you for your response. We clarify that the revised Appendix is included in the Supplementary Material, which is accessible using the associated option below the paper's abstract on the OpenReview page. Thank you.

---

### Official Review · Reviewer_uFrD · 2025-10-27

**Soundness:** 2
**Presentation:** 3
**Contribution:** 1
**Rating:** 2
**Confidence:** 5

**Summary:**

This paper revisits the role of geometry-grounded vision backbones (e.g., VGGT) versus visual-only ones (e.g., DINOv2/v3) in semantic distillation for radiance fields. It examines three key questions: Do geometry-grounded features contain richer spatial information? Do they improve semantic object localization? And do they enable more accurate radiance field inversion (pose estimation)?
The authors propose SPINE, a novel inversion framework using semantic features to estimate camera poses without initialization, followed by photometric refinement. Surprisingly, results show that geometry-grounded features do not outperform visual-only ones on localization or inversion, though they contain more geometric detail.

**Strengths:**

1. This is the first systematic analysis comparing visual-only vs. visual-geometry semantic embeddings in radiance fields, a gap overlooked by existing works like LERF, CLIP-NeRF, or DFF.
2. The study spans multiple datasets (LERF, 3D-OVS, Robotics), two radiance field types (NeRF, GS), and multiple metrics (GFF, SSIM, PSNR, LPIPS, SE(3) error).

**Weaknesses:**

1. The description of SPINE’s inverse model is conceptually clear but lacks comparative or sensitivity analyses: How critical are the semantic embeddings vs. the photometric refinement? How do different backbone dimensions (e.g., CLIP 512 vs 768) affect inversion? What is the runtime/efficiency cost of SPINE relative to baseline pose estimators?

2. The paper concludes that geometry-grounded semantics hurt versatility but offers no concrete explanation beyond “supervised inductive bias.”

3. Semantic localization results are mostly relative (DINO vs VGGT). It would be more informative to include absolute comparisons against: CLIP-only localization (as in LERF), Geometry-only cues (depth or SDF features).

4. The term "semantic embedding" is misleading in L161. VGGT is not trained with semantic supervision, and I don't believe its intermediate features can be called semantic embedding.

5. The core idea, comparing geometry-grounded vs. visual-only features in semantic radiance fields, is primarily evaluative.
SPINE, the only algorithmic contribution, is a straightforward combination of: a shallow MLP mapping semantic embeddings to poses, and standard PnP-based refinement. Both steps are well established in prior literature (e.g., iNeRF [Yen-Chen 2021], CatNIPS [Chen 2024], and Splat-NAV [Chen 2025]). SPINE’s novelty lies mostly in not requiring an initial guess, but the authors never prove that it consistently converges without one — the results appear scene-specific and qualitative.
Overall, the paper feels like an empirical case study rather than a fundamentally new framework or theoretical advance.

6. Mathematical sections contain excessive exposition of standard concepts (e.g., SVD, Sobel operator). Section 4 devotes a full paragraph to PCA projection details, which are trivial and distract from the main analysis.

**Questions:**

1. Could you provide any intuition or analysis for why geometry grounding degrades downstream task performance?
2. Does SPINE generalize across unseen scenes, or is it trained per scene like a NeRF?
3. You conclude that “visual-only features offer greater versatility.” How general is this conclusion? Do you believe this holds for all geometry-grounded models (e.g., MVDream, Depth-DINO, GeoCLR), or only VGGT? What would you recommend for future work — redesigning geometry-grounding losses or simply abandoning geometry-grounded semantics for robotics applications?

---

> ### Author Response · Authors · 2025-11-26
> **Response to Reviewer uFrD (Part I)**
>
> Thank you for recognizing the value of the research questions
> considered in our work. We are grateful for your valuable feedback,
> which we found to be really helpful and constructive. We group our
> response into the following sections:
>
> **SPINE.** We have revised the paper to provide a detailed discussion
> of SPINE, analyzing the performance of the coarse and fine inversion
> procedures in SPINE. We highlight that the novel coarse inversion
> phase of SPINE enables it to achieve much higher success rates
> compared to the baseline methods in both Gaussian Splatting
> and NeRF scenes. We also show that the fine inversion phase
> significantly improves the accuracy of the solution returned by
> SPINE. Prior work [12] has examined the effects of different
> backbone dimensions, without finding any significant differences. In
> preliminary experiments, we did not find any significant differences.
> After a one-time setup pass, SPINE runs at about 2 Hz, essentially
> as fast as the method in [ 10 ] and much faster than iNeRF, which
> runs at 0.05 Hz for 100 optimization steps.
>
> **Discussion of the Content of Semantic Features.** We have revised
> our paper to more extensively probe the content of the semantic
> features of these models in Section 5. We highlight a few major
> points here. Across all datasets, we observe that visual-semantic
> features (e.g., from DINO) capture higher-resolution object-level
> semantic detail; however, these features do not encapsulate high-
> fidelity object morphology. In contrast, geometry-supervised encoders
> give up entity-level semantic information to attend to structural
> components of the scene. Importantly, our findings suggest that
> more significant finetuning generally leads to a greater shift in the
> attention of these pretrained encoders from object-level semantic
> information to the objects’ geometries. Further, these results suggest
> that visual-semantic encoders would excel at tasks that do not require
> knowledge of the precise geometry of objects, like many pick-and-
> place tasks in robot manipulation; however, dexterous manipulation
> tasks, such as multi-finger prehensile tasks, might benefit from
> task-supervised vision encoders, such as the geometry-grounded
> encoders discussed in our work. We can attribute the degradation in
> the semantic knowledge of the finetuned encoders to catastrophic
> forgetting, where the encoders lose prior knowledge of object-level
> semantic information as finetuning progresses. As demonstrated in
> prior work, our findings suggest finetuning with low-rank adaptation
> (LoRA) could be essential in preserving the semantic knowledge
> of vision encoders. Further, finetuned encoders tend to overfit to
> the task, which can be addressed through multi-objective training
> frameworks using loss functions with a task-focused component and
> a task-agnostic component to promote generalization.

---

> ### Author Response · Authors · 2025-11-26
> **Response to Reviewer uFrD (Part II)**
>
> **Semantic Localization.** We clarify that we apply standard practice
> in our semantic localization, which essentially matches the pipeline
> used in LERF. Specifically, LERF uses CLIP augmented with DINO
> for localization, which is exactly one of the settings evaluated in
> our paper. Note that CLIP is not used in isolation, rather, it provides
> open-vocabulary capabilities for DINO. Likewise, we do not consider
> geometry-only cues in our work because we only assume access
> to a monocular camera. In fact, all the radiance fields in our work
> were trained without any depth information.
>
> **Semantic embedding.** We clarify that VGGT finetunes DINOv2
> for 3D scene reconstruction, hence its association with the term
> semantic embeddings. The resulting VGGT pipeline extract semantic
> features from input images to generate 3D scenes along with the
> corresponding camera intrinsic and extrinsic parameters. In essence,
> we use the term “semantic embedding” in a broader sense.
> Contribution. We apologize for the confusion on the core contribu-
> tion of our work. We have revised the abstract and introduction of our
> paper to clarify this point. Our main contribution is a systematic study
> of pretrained semantics to uncover the salient components of semantic
> features of prominent vision encoders in robotics. We consider visual-
> semantic (e.g., DINO) and geometry-grounded vision encoders, as
> representative encoders, given the important role played by geometry
> and semantics in robotics tasks. An avid reader might ask why this
> study is important or useful. We emphasize that despite the ubiquity
> of pretrained vision encoders, a striking lack of consensus exists on
> the characteristics of pretrained semantics from these encoders. For
> example, some prior works [ 1] claim that (task-specific) pretrained
> encoders outperform visual-semantic encoders (CLIP/DINO) or that
> fine-tuning vision encoders hurts performance [2, 3], while other
> works [ 4] claim the exact opposite. These contradictory findings pose
> a major challenge to researchers and practitioners alike. Our study
> seeks to provide insights that address these divisions, guided by the
> intuition that the secret to understanding these seemingly inconsistent
> claims lies in the internal content of the semantic features of these
> encoders.
>
> Notably, in robotics, there is a prominent lack of research
> examining the relative composition of semantic features across
> vision encoders to inform better integration of these encoders into
> robotics pipelines, with a few exceptions from the computer vision
> community, such as DINO. Through our study, we reveal that visual-
> semantic encoders focus on preserving object/part-level semantic
> information that distinguishes between different classes of objects,
> while task-specific (geometry-grounded) finetuned encoders may
> discard this semantic information in favor of structural information
> that emphasize fine-grained edges, corners, and other spatial details.
> Our findings suggest that visual-semantic encoders (such as DINO)
> would outperform task-specific encoders (such as DUSt3R, MASt3R,
> VGGT, etc.) in generalist robot manipulation across a broad range
> of tasks, as observed in prior work [ 4, 3]. On the other hand,
> task-specific encoders would likely outperform visual-semantic
> encoders on dexterous robot manipulation tasks, given sufficient
> training data coverage, as observed in existing work [ 1]. Further,
> our findings highlight that finetuning might degrade generalization
> by replacing more generalizable feature content with task-specific
> information, which resolves the discrepancies in the results of
> prior work, e.g., [ 2, 4 , 3]. We would also like to reiterate that our
> works provides actionable insights to researchers and practitioners
> alike. Hence, our work is well-aligned with the overall goal of
> ICLR: “Submissions bring value to the ICLR community when they
> convincingly demonstrate new, relevant, impactful knowledge (incl.,
> empirical, theoretical, for practitioners, etc).”
>
> We would also like to clarify that SPINE is a secondary
> contribution in our work. Nonetheless, SPINE turns out to be a
> significant contribution that addresses one of the major drawbacks
> of existing radiance field inversion methods, by eliminating the
> need for an initial guess. We agree that SPINE may be considered
> a straightforward solution. However, the term straightforward can
> be misleading. We would like to disambiguate between simple
> solutions and obvious solutions, which are both often interpreted as
> straightforward. We acknowledge that SPINE is a simple, effective
> solution. However, if this solution was obvious, existing methods
> would have implemented it to address their weaknesses, given
> the significance of these weaknesses. However, no prior work
> has developed a solution to eliminate reliance on an initial guess,
> underscoring the importance of our contribution.

---

> ### Author Response · Authors · 2025-11-26
> **Response to Reviewer uFrD (Part III)**
>
> **Mathematical Sections.** We have revised the paper to move details
> about SVD and PCA to the appendix.
> SPINE trained per Scene? Like all camera relocalization and
> radiance field inversion methods, SPINE is trained per scene.
> However, SPINE is still applicable to real robotic deployments,
> as demonstrated in prior work [10].
>
> **Other Geometry-Grounded Models.** We have included additional
> experiments with DUSt3R, MASt3R, and MUSt3R to broaden the
> scope of spatially grounded models. In Section 4, we provide a
> detailed discussion of the semantic content of these models. We
> find that our conclusions on VGGT generalize to these models. For
> example, these model attend to structural information in their input
> images, discarding object-level semantic information for more fine-
> grained geometric details, which could hurt their performance in
> tasks that more heavily depend on object-level semantic information.
>
> Thank you for your feedback. We hope our revisions and response
> has addressed your concerns about the paper. We would be glad if
> you could consider updating your score to reflect the revision.
>
> [1] Cheng Chi, Zhenjia Xu, Siyuan Feng, Eric Cousineau, Yilun Du, Benjamin Burchfiel, Russ Tedrake, and Shuran Song. Diffusion policy: Visuomotor policy learning via action diffusion. The International Journal of Robotics Research, 44(10-11):1684–1704, 2025.
>
> [2] Siddharth Karamcheti, Suraj Nair, Ashwin Balakrishna, Percy Liang, Thomas Kollar, and Dorsa Sadigh. Prismatic VLMs: Investigating the Design Space of Visually-Conditioned Language Models. In International Conference on Machine Learning (ICML), volume 235, pages 11197–11210. PMLR, 2024.
>
> [3] Huang Huang, Fangchen Liu, Letian Fu, Tingfan Wu, Mustafa Mukadam, Jitendra Malik, Ken Goldberg, and Pieter Abbeel. Otter: A vision-language-action model with text-aware visual feature extraction. arXiv preprint arXiv:2503.03734, 2025.
>
> [4] Moo Jin Kim, Karl Pertsch, Siddharth Karamcheti, Ted Xiao, Ashwin Balakrishna, Suraj Nair, Rafael Rafailov, Ethan P Foster, Pannag R Sanketi, Quan Vuong, et al. Openvla: An open-source vision-language-action model. In Conference on Robot Learning, pages 2679–2713. PMLR, 2025.
>
> [5] Alex Kendall, Matthew Grimes, and Roberto Cipolla. Posenet: A convolutional network for real-time 6-dof camera relocalization. In Proceedings of the IEEE international conference on computer vision, pages 2938–2946, 2015.
>
> [6] Fei Xue, Xin Wu, Shaojun Cai, and Junqiu Wang. Learning multi- view camera relocalization with graph neural networks. In 2020 IEEE/CVF Conference on Computer Vision and Pattern Recognition (CVPR), pages 11372–11381. IEEE, 2020.
>
> [7] Qunjie Zhou, Maxim Maximov, Or Litany, and Laura Leal-Taix ´e. The nerfect match: Exploring nerf features for visual localization. In European Conference on Computer Vision, pages 108–127. Springer, 2024.
>
> [8] Vincent Leroy, Yohann Cabon, and Jerome Revaud. Grounding image matching innbsp;3d withnbsp;mast3r. In Computer Vision – ECCV 2024: 18th European Conference, Milan, Italy, September 29–October 4, 2024, Proceedings, Part LXXII, page 71–91, Berlin, Heidelberg, 2024. Springer-Verlag. ISBN 978-3-031-73219-5. doi: 10.1007/978-3-031-73220-1 5. URL https://doi.org/10.1007/ 978-3-031-73220-1 5.
>
> [9] Jianyuan Wang, Minghao Chen, Nikita Karaev, Andrea Vedaldi, Christian Rupprecht, and David Novotny. Vggt: Visual geometry grounded transformer. In Proceedings of the Computer Vision and Pattern Recognition Conference, pages 5294–5306, 2025.
>
> [10] Timothy Chen, Ola Shorinwa, Joseph Bruno, Aiden Swann, Javier
> Yu, Weijia Zeng, Keiko Nagami, Philip Dames, and Mac Schwager.
> Splat-nav: Safe real-time robot navigation in gaussian splatting
> maps. IEEE Transactions on Robotics, 2025.
>
> [11] Justin Kerr, Chung Min Kim, Ken Goldberg, Angjoo Kanazawa,
> and Matthew Tancik. Lerf: Language embedded radiance fields. In
> Proceedings of the IEEE/CVF international conference on computer
> vision, pages 19729–19739, 2023.
>
> [12] William Shen, Ge Yang, Alan Yu, Jansen Wong, Leslie Pack
> Kaelbling, and Phillip Isola. Distilled feature fields enable few-shot
> language-guided manipulation. arXiv preprint arXiv:2308.07931,
> 2023.

---

### Official Review · Reviewer_mf1Y · 2025-10-31

**Soundness:** 3
**Presentation:** 3
**Contribution:** 2
**Rating:** 4
**Confidence:** 4

**Summary:**

This paper investigates whether geometry-grounded vision backbones, specifically VGGT, can provide advantages over purely visual backbones such as DINOv2/DINOv3 when distilled into radiance fields for robotics-centric tasks. The authors evaluate three downstream capabilities: semantic content fidelity, open-vocabulary object localization, and radiance-field inversion. The study concludes that geometry-grounded features exhibit sharper geometric structure but do not improve semantic localization performance and even degrade pose inversion accuracy. The paper additionally introduces SPINE, a scene-specific inversion module that leverages semantic cues to recover camera pose without initialization.

The central message is that geometry-enhanced features do not necessarily translate into broader utility in 3D-aware semantic radiance fields, and that purely visual embeddings remain more versatile for downstream tasks.

**Strengths:**

1. Addresses a timely and relevant question regarding the real-world value of geometry-grounded semantics for 3D robotics perception.

2. Provides systematic comparisons across multiple semantic backbones and tasks.

3. Experimental setup is generally clear and the negative results are informative for the community.

**Weaknesses:**

1. **Novelty is limited**
   - The core technical pipeline largely reuses existing radiance-field semantic distillation approaches, with the primary modification being the substitution of pretrained feature sources.
   - SPINE follows a standard design (semantic prior + photometric refinement + PnP/RANSAC) and is trained per scene, further reducing novelty in system design.

2. **Scope of geometry-grounded models is insufficient**
   - Only VGGT is evaluated. Contemporary spatially grounded models such as DUSt3R, MASt3R, CroCo, or other geometric transformer variants are omitted.
   - The conclusion that geometry-grounding harms versatility is based on a narrow model sample and may not generalize.

3. **Lack of mechanistic insight**
   - The paper observes performance degradation with geometry-grounded features but does not provide clear hypotheses or analysis explaining why geometry hurts semantic versatility.
   - Without deeper investigation, the conclusions may appear anecdotal.

4. **Incomplete evaluation of scalability and practicality**
   - SPINE is trained per scene, similar to NeRF-style pipelines, raising concerns about scalability in real robotic deployments.
   - No demonstration on larger-scale scenes, dynamic environments, or cross-scene generalization.

5. **Overall maturity not yet sufficient**
   - Although the question is valuable, the current implementation resembles an exploratory empirical study rather than a fully developed methodology.
   - Lack of ablations on distillation design (e.g., shared hashgrids, separate semantic heads, language-only vs geometry-only distillation) limits interpretability.

**Questions:**

1. Would the observed trend hold for other geometry-grounded models such as DUSt3R, MASt3R, or other spatial transformers?
2. Can the authors provide more in-depth analysis explaining why geometry-grounding compromises semantic versatility? For example, changes in embedding smoothness, gradient stability, or photometric consistency?
3. Is SPINE capable of generalizing across scenes, or can it be made scene-agnostic? If not, how do the authors envision scaling it in real robotic deployments?
4. Could separate encodings rather than shared hash-grids improve geometry-grounded feature distillation?
5. Do any tasks exist where the geometry-grounded semantic fields *do* offer measurable benefits?

---

> ### Author Response · Authors · 2025-11-26
> **Response to Reviewer mf1Y (Part I)**
>
> Thank you for recognizing the value of our work. We are grateful
> for your valuable feedback, which we found to be really helpful and
> constructive. We group our response into the following sections:
>
> **Novelty.** We apologize for the confusion on the core contribution
> of our work. We have revised the abstract and introduction of our
> paper to clarify this point. Our main contribution is a systematic
> study of pretrained semantics to uncover the salient components of
> semantic features of robotics vision encoders. We consider visual-
> semantic (e.g., DINO) and geometry-grounded vision encoders, as
> representative examples, given the important role played by geometry
> and semantics in robotics tasks. An avid reader might ask why this
> study is important or useful. We emphasize that despite the ubiquity
> of pretrained vision encoders, a striking lack of consensus exists on
> the characteristics of pretrained semantics from these encoders. For
> example, some prior works [ 1] claim that (task-specific) pretrained
> encoders outperform visual-semantic encoders (CLIP/DINO) or that
> fine-tuning vision encoders hurts performance [2, 3], while other
> works [ 4] claim the exact opposite. These contradictory findings pose
> a major challenge to researchers and practitioners alike. Our study
> seeks to provide insights that address these divisions, guided by the
> intuition that the secret to understanding these seemingly inconsistent
> claims lies in the internal content of the semantic features of these
> encoders.
>
> Notably, in robotics, there is a prominent lack of research
> examining the relative composition of semantic features across
> vision encoders to inform better integration of these encoders into
> robotics pipelines, with a few exceptions from the computer vision
> community, such as DINO. Through our study, we reveal that visual-
> semantic encoders focus on preserving object/part-level semantic
> information that distinguishes between different classes of objects,
> while task-specific (geometry-grounded) finetuned encoders may
> discard this semantic information in favor of structural information
> that emphasize fine-grained edges, corners, and other spatial details.
> Our findings suggest that visual-semantic encoders (such as DINO)
> would outperform task-specific encoders (such as DUSt3R, MASt3R,
> VGGT, etc.) in generalist robot manipulation across a broad range
> of tasks, as observed in prior work [ 4, 3]. On the other hand,
> task-specific encoders would likely outperform visual-semantic
> encoders on dexterous robot manipulation tasks, given sufficient
> training data coverage, as observed in existing work [ 1]. Further,
> our findings highlight that finetuning might degrade generalization
> by replacing more generalizable feature content with task-specific
> information, which resolves the discrepancies in the results of
> prior work, e.g., [ 2, 4, 3]. We would also like to reiterate that our
> works provides actionable insights to researchers and practitioners
> alike. Hence, our work is well-aligned with the overall goal of
> ICLR: “Submissions bring value to the ICLR community when they
> convincingly demonstrate new, relevant, impactful knowledge (incl.,
> empirical, theoretical, for practitioners, etc).”

---

> > ### Author Response · Authors · 2025-11-26
> > **Response to Reviewer mf1Y (Part II)**
> >
> > **Scope of geometry-grounded models.** We have included additional
> > experiments with DUSt3R, MASt3R, and MUSt3R to broaden the
> > scope of spatially grounded models. In Section 4, we provide a
> > detailed discussion of the semantic content of these models. We
> > find that our conclusions on VGGT generalize to these models. For
> > example, these models attend to structural information in their input
> > images, discarding object-level semantic information for more fine-
> > grained geometric details, which could hurt their performance in
> > tasks that more heavily depend on object-level semantic information.
> > Mechanistic Insight. We have revised our paper to more extensively
> > probe the content of the semantic features of these models in
> > Section 5. We highlight a few major points here. Across all datasets,
> > we observe that visual-semantic features (e.g., from DINO) capture
> > higher-resolution object-level semantic detail; however, these features
> > do not encapsulate high-fidelity object morphology. In contrast,
> > geometry-supervised encoders give up entity-level semantic informa-
> > tion to attend to structural components of the scene. Importantly, our
> > findings suggest that more significant finetuning generally leads to a
> > greater shift in the attention of these pretrained encoders from object-
> > level semantic information to the objects’ geometries. Further, these
> > results suggest that visual-semantic encoders would excel at tasks
> > that do not require knowledge of the precise geometry of objects,
> > like many pick-and-place tasks in robot manipulation; however,
> > dexterous manipulation tasks, such as multi-finger prehensile tasks,
> > might benefit from task-supervised vision encoders, such as the
> > geometry-grounded encoders discussed in our work. We can attribute
> > the degradation in the semantic knowledge of the finetuned encoders
> > to catastrophic forgetting, where the encoders lose prior knowledge
> > of object-level semantic information as finetuning progresses. As
> > demonstrated in prior work, our findings suggest finetuning with
> > low-rank adaptation (LoRA) could be essential in preserving the
> > semantic knowledge of vision encoders. Further, finetuned encoders
> > tend to overfit to the task, which can be addressed through multi-
> > objective training frameworks using loss functions with a task-
> > focused component and a task-agnostic component to promote
> > generalization.
> >
> > **Evaluation of SPINE.** We clarify that radiance field inversion is
> > a well-established research area [10 ]. Our work follows standard
> > practice in this research area, evaluating SPINE across many
> > benchmark scenes. We have revised our discussion of radiance
> > field inversion to better discuss the nuances of this task compared
> > to other related tasks. We emphasize that although radiance field
> > inversion is related to a number of other problems, e.g., camera
> > relocalization [5, 6 , 7] and 3D scene reconstruction [8, 9], a few
> > key characteristics distinguish radiance field inversion from these
> > tasks. Specifically, camera relocalization methods (e.g., [ 5]) generally
> > require lots of training data (on the order of thousands to tens of
> > thousands). In contrast, radiance field inversion typically utilizes
> > training data with fewer than a hundred (or a few hundred) samples,
> > a  scale impractical for camera relocalization methods. Likewise,
> > radiance field inversion methods computes poses in a global reference
> > frame, unlike 3D scene reconstruction methods. Although SPINE
> > is trained per scene like all camera relocalization and radiance
> > field inversion methods, SPINE is still applicable to real robotic
> > deployments, as demonstrated in prior work [10].
> >
> > We would also like to clarify that SPINE is a secondary
> > contribution in our work. Nonetheless, SPINE turns out to be a
> > significant contribution that addresses one of the major drawbacks
> > of existing radiance field inversion methods, by eliminating the need
> > for an initial guess. If this solution was obvious, existing methods
> > would have implemented it to address their weaknesses. However,
> > no prior work has developed a solution to eliminate reliance on an
> > initial guess, underscoring the importance of our contribution.
> >
> > **Ablations.** We have revised our paper to better highlight its main
> > contributions. Our paper focuses on probing pretrained semantics
> > to reveal actionable insights into their internal content. We do not
> > run ablations on distillation designs for two reasons. Prior work
> > has evaluated alternative distillation designs, which we summarize
> > here. The shared hashgrids are necessary to enable joint learning of
> > CLIP in combination with other semantic features, as discussed in
> > Section 4. Moreover, we emphasize that in line with prior work [ 11],
> > we use CLIP to augment the capabilities of the other vision encoders
> > to support open-vocabulary interaction.
> >
> > Thank you for your feedback. We hope our revisions and response
> > has addressed your concerns about the paper. We would be glad if
> > you could consider updating your score to reflect the revision.

---

> ### Author Response · Authors · 2025-11-26
> **Response to Reviewer mf1Y (Part III)**
>
> [1] Cheng Chi, Zhenjia Xu, Siyuan Feng, Eric Cousineau, Yilun Du, Benjamin Burchfiel, Russ Tedrake, and Shuran Song. Diffusion policy: Visuomotor policy learning via action diffusion. The International Journal of Robotics Research, 44(10-11):1684–1704, 2025.
>
> [2] Siddharth Karamcheti, Suraj Nair, Ashwin Balakrishna, Percy Liang, Thomas Kollar, and Dorsa Sadigh. Prismatic VLMs: Investigating the Design Space of Visually-Conditioned Language Models. In International Conference on Machine Learning (ICML), volume 235, pages 11197–11210. PMLR, 2024.
>
> [3] Huang Huang, Fangchen Liu, Letian Fu, Tingfan Wu, Mustafa Mukadam, Jitendra Malik, Ken Goldberg, and Pieter Abbeel. Otter: A vision-language-action model with text-aware visual feature extraction. arXiv preprint arXiv:2503.03734, 2025.
>
> [4] Moo Jin Kim, Karl Pertsch, Siddharth Karamcheti, Ted Xiao, Ashwin Balakrishna, Suraj Nair, Rafael Rafailov, Ethan P Foster, Pannag R Sanketi, Quan Vuong, et al. Openvla: An open-source vision-language-action model. In Conference on Robot Learning, pages 2679–2713. PMLR, 2025.
>
> [5] Alex Kendall, Matthew Grimes, and Roberto Cipolla. Posenet: A convolutional network for real-time 6-dof camera relocalization. In Proceedings of the IEEE international conference on computer vision, pages 2938–2946, 2015.
>
> [6] Fei Xue, Xin Wu, Shaojun Cai, and Junqiu Wang. Learning multi- view camera relocalization with graph neural networks. In 2020 IEEE/CVF Conference on Computer Vision and Pattern Recognition (CVPR), pages 11372–11381. IEEE, 2020.
>
> [7] Qunjie Zhou, Maxim Maximov, Or Litany, and Laura Leal-Taix ´e. The nerfect match: Exploring nerf features for visual localization. In European Conference on Computer Vision, pages 108–127. Springer, 2024.
>
> [8] Vincent Leroy, Yohann Cabon, and Jerome Revaud. Grounding image matching innbsp;3d withnbsp;mast3r. In Computer Vision – ECCV 2024: 18th European Conference, Milan, Italy, September 29–October 4, 2024, Proceedings, Part LXXII, page 71–91, Berlin, Heidelberg, 2024. Springer-Verlag. ISBN 978-3-031-73219-5. doi: 10.1007/978-3-031-73220-1 5. URL https://doi.org/10.1007/ 978-3-031-73220-1 5.
>
> [9] Jianyuan Wang, Minghao Chen, Nikita Karaev, Andrea Vedaldi, Christian Rupprecht, and David Novotny. Vggt: Visual geometry grounded transformer. In Proceedings of the Computer Vision and Pattern Recognition Conference, pages 5294–5306, 2025.
>
> [10] Timothy Chen, Ola Shorinwa, Joseph Bruno, Aiden Swann, Javier Yu, Weijia Zeng, Keiko Nagami, Philip Dames, and Mac Schwager. Splat-nav: Safe real-time robot navigation in gaussian splatting maps. IEEE Transactions on Robotics, 2025.
>
> [11] Justin Kerr, Chung Min Kim, Ken Goldberg, Angjoo Kanazawa, and Matthew Tancik. Lerf: Language embedded radiance fields. In Proceedings of the IEEE/CVF international conference on computer vision, pages 19729–19739, 2023.
>
> [12] William Shen, Ge Yang, Alan Yu, Jansen Wong, Leslie Pack Kaelbling, and Phillip Isola. Distilled feature fields enable few-shot language-guided manipulation. arXiv preprint arXiv:2308.07931, 2023.

---

### Official Review · Reviewer_HjDG · 2025-11-02

**Soundness:** 2
**Presentation:** 2
**Contribution:** 1
**Rating:** 2
**Confidence:** 4

**Summary:**

In this paper, the authors explore the effectiveness of vision-only and visual-geometry features on downstream tasks including edge computation, object localization, etc.

Additionally, the authors propose a SPINE framework for inverting radiance fields. More precisely, SPINE predicts poses directly from learned image features and then refines the poses by solving a PnP problem.


Experiments on public datasets demonstrate that (1) visual-geometry features contain higher fidelity spatial content than visual-only features; (2) both features give close performance on semantic object localization; (3) visual-only features give higher accuracy radiance field inversion.

**Strengths:**

1.	Interesting topic. Recently, distilling semantic knowledge from foundation models (e.g. CLIP) into NeRFs or Gaussian Splatting is a hot research topic as also mentioned in the introduction. However, comparisons of the effectiveness of the visual-only features (e.g. DINO v2/v3) and visual-geometry features (e.g. VGGT) on different tasks are still open problems. Therefore, I believe the topic of this paper is interesting and the conclusions obtained from the experiments are useful.

2.	The paper is well-organized.

3.	The experiments, although conducted on a limited number of datasets, are relatively convincing.

**Weaknesses:**

1.	Limited contribution. The main contribution of this paper is the comparison of the effectiveness of visual-only and visual-geometry features on downstream tasks. Although it is interesting and useful, the contribution is not enough as an ICLR paper.

2.	The second contribution of the paper is the SPINE framework for inverting radiance fields. However, the SPINE is essentially a pipeline for end-to-end relocalization task which takes features as input and predicts the 6-DoF camera poses. There are many works in this area [R1, R2, R3]. It would be better to give a discussion of this task.

[R1] PoseNet: A Convolutional Network for Real-Time 6-DOF Camera Relocalization, Kendall  et al., ICCV 2015.

[R2] Learning multi-view camera relocalization with graph neural networks, xue et al., ICCV 2020.

[R3] The NeRFect Match: Exploring NeRF Features for Visual Localization, zhou et al., ECCV 2024.

**Questions:**

It seems like some details of the method are missing, which causes the low readability, for example:

1.	The details of f_l, f_s.

2.	How to the match before PnP for pose estimation.

---

> ### Author Response · Authors · 2025-11-26
> **Response to Reviewer HjDG (Part I)**
>
> Thank you for recognizing the value of the research questions
> considered in our work. We are grateful for your valuable feedback,
> which we found to be really helpful and constructive. We group our
> response into the following sections:
>
> **Limited contribution.** We apologize for the confusion on the
> core contribution of our work. We have revised the abstract and
> introduction of our paper to clarify this point. Our main contribution
> is a systematic study of pretrained semantics to uncover the salient
> components of semantic features of robotics vision encoders. We
> consider visual-semantic (e.g., DINO) and geometry-grounded vision
> encoders, as representative examples, given the important role played
> by geometry and semantics in robotics tasks. An avid reader might
> ask why this study is important or useful. We emphasize that
> despite the ubiquity of pretrained vision encoders, a striking lack
> of consensus exists on the characteristics of pretrained semantics
> from these encoders. For example, some prior works [1] claim
> that (task-specific) pretrained encoders outperform visual-semantic
> encoders (CLIP/DINO) or that fine-tuning vision encoders hurts
> performance [ 2, 3 ], while other works [ 4] claim the exact opposite.
> These contradictory findings pose a major challenge to researchers
> and practitioners alike. Our study seeks to provide insights that
> address these divisions, guided by the intuition that the secret to
> understanding these seemingly inconsistent claims lies in the internal
> content of the semantic features of these encoders.

---

> ### Author Response · Authors · 2025-11-26
> **Response to Reviewer HjDG (Part II)**
>
> Notably, in robotics, there is a prominent lack of research
> examining the relative composition of semantic features across
> vision encoders to inform better integration of these encoders into
> robotics pipelines, with a few exceptions from the computer vision
> community, such as DINO. Through our study, we reveal that visual-
> semantic encoders focus on preserving object/part-level semantic
> information that distinguishes between different classes of objects,
> while task-specific (geometry-grounded) finetuned encoders may
> discard this semantic information in favor of structural information
> that emphasize fine-grained edges, corners, and other spatial details.
> Our findings suggest that visual-semantic encoders (such as DINO)
> would outperform task-specific encoders (such as DUSt3R, MASt3R,
> VGGT, etc.) in generalist robot manipulation across a broad range
> of tasks, as observed in prior work [ 4, 3]. On the other hand,
> task-specific encoders would likely outperform visual-semantic
> encoders on dexterous robot manipulation tasks, given sufficient
> training data coverage, as observed in existing work [ 1]. Further,
> our findings highlight that finetuning might degrade generalization
> by replacing more generalizable feature content with task-specific
> information, which resolves the discrepancies in the results of
> prior work, e.g., [ 2, 4, 3].
> We would also like to reiterate that our
> works provides actionable insights to researchers and practitioners
> alike. Hence, our work is well-aligned with the overall goal of
> ICLR: “Submissions bring value to the ICLR community when they
> convincingly demonstrate new, relevant, impactful knowledge (incl.,
> empirical, theoretical, for practitioners, etc).”
>
>
> **Relationship between SPINE and Camera Relocalization.** We
> have described radiance field inversion more clearly in Section 4
> in the revised paper. We emphasize that although radiance field
> inversion is related to a number of other problems, e.g., camera
> relocalization [5, 6 , 7] and 3D scene reconstruction [8, 9], a few
> key characteristics distinguish radiance field inversion from these
> tasks. Specifically, camera relocalization methods (e.g., [ 5]) generally
> equire lots of training data (on the order of thousands to tens of
> thousands). In contrast, radiance field inversion typically utilizes
> training data with fewer than a hundred (or a few hundred) samples,
> a scale impractical for camera relocalization methods. Likewise,
> radiance field inversion methods computes poses in a global reference
> frame, unlike 3D scene reconstruction methods. We would also
> like to clarify that SPINE is a secondary contribution in our work.
> Nonetheless, SPINE turns out to be a significant contribution that
> addresses one of the major drawbacks of existing radiance field
> inversion methods, by eliminating the need for an initial guess. If
> this solution was obvious, existing methods would have implemented
> it to address their weaknesses. However, no prior work has developed
> a solution to eliminate reliance on an initial guess, underscoring the
> importance of our contribution.
>
>
> **Missing Details.**  We explain the details of fl and fs, clarifying that
> these fields are parameterized by an MLP which map 3D points
> to semantic features. We provide additional details on the size of
> the MLPs in Appendix A.3. Further, we have revised the paper to
> indicate that we use SIFT for image feature extraction and matching
> in PnP.
>
> Thank you for your feedback. We hope our revisions and response
> has addressed your concerns about the paper. We would be glad if
> you could consider updating your score to reflect the revision.

---

> ### Author Response · Authors · 2025-11-26
> **Response to Reviewer HjDG (Part III)**
>
> [1] Cheng Chi, Zhenjia Xu, Siyuan Feng, Eric Cousineau, Yilun Du, Benjamin Burchfiel, Russ Tedrake, and Shuran Song. Diffusion policy: Visuomotor policy learning via action diffusion. The International Journal of Robotics Research, 44(10-11):1684–1704, 2025.
>
> [2] Siddharth Karamcheti, Suraj Nair, Ashwin Balakrishna, Percy Liang, Thomas Kollar, and Dorsa Sadigh. Prismatic VLMs: Investigating the Design Space of Visually-Conditioned Language Models. In International Conference on Machine Learning (ICML), volume 235, pages 11197–11210. PMLR, 2024.
>
> [3] Huang Huang, Fangchen Liu, Letian Fu, Tingfan Wu, Mustafa Mukadam, Jitendra Malik, Ken Goldberg, and Pieter Abbeel. Otter: A vision-language-action model with text-aware visual feature extraction. arXiv preprint arXiv:2503.03734, 2025.
>
> [4] Moo Jin Kim, Karl Pertsch, Siddharth Karamcheti, Ted Xiao, Ashwin Balakrishna, Suraj Nair, Rafael Rafailov, Ethan P Foster, Pannag R Sanketi, Quan Vuong, et al. Openvla: An open-source vision-language-action model. In Conference on Robot Learning, pages 2679–2713. PMLR, 2025.
>
> [5] Alex Kendall, Matthew Grimes, and Roberto Cipolla. Posenet: A convolutional network for real-time 6-dof camera relocalization. In Proceedings of the IEEE international conference on computer vision, pages 2938–2946, 2015.
>
> [6] Fei Xue, Xin Wu, Shaojun Cai, and Junqiu Wang. Learning multi- view camera relocalization with graph neural networks. In 2020 IEEE/CVF Conference on Computer Vision and Pattern Recognition (CVPR), pages 11372–11381. IEEE, 2020.
>
> [7] Qunjie Zhou, Maxim Maximov, Or Litany, and Laura Leal-Taix ´e. The nerfect match: Exploring nerf features for visual localization. In European Conference on Computer Vision, pages 108–127. Springer, 2024.
>
> [8] Vincent Leroy, Yohann Cabon, and Jerome Revaud. Grounding image matching innbsp;3d withnbsp;mast3r. In Computer Vision – ECCV 2024: 18th European Conference, Milan, Italy, September 29–October 4, 2024, Proceedings, Part LXXII, page 71–91, Berlin, Heidelberg, 2024. Springer-Verlag. ISBN 978-3-031-73219-5. doi: 10.1007/978-3-031-73220-1 5. URL https://doi.org/10.1007/ 978-3-031-73220-1 5.
>
> [9] Jianyuan Wang, Minghao Chen, Nikita Karaev, Andrea Vedaldi, Christian Rupprecht, and David Novotny. Vggt: Visual geometry grounded transformer. In Proceedings of the Computer Vision and Pattern Recognition Conference, pages 5294–5306, 2025.
>
> [10] Timothy Chen, Ola Shorinwa, Joseph Bruno, Aiden Swann, Javier Yu, Weijia Zeng, Keiko Nagami, Philip Dames, and Mac Schwager. Splat-nav: Safe real-time robot navigation in gaussian splatting maps. IEEE Transactions on Robotics, 2025.
>
> [11] Justin Kerr, Chung Min Kim, Ken Goldberg, Angjoo Kanazawa, and Matthew Tancik. Lerf: Language embedded radiance fields. In Proceedings of the IEEE/CVF international conference on computer vision, pages 19729–19739, 2023.
>
> [12] William Shen, Ge Yang, Alan Yu, Jansen Wong, Leslie Pack Kaelbling, and Phillip Isola. Distilled feature fields enable few-shot language-guided manipulation. arXiv preprint arXiv:2308.07931, 2023.

---

### Meta-Review · Area_Chair_SwuQ · 2026-01-06

**Summary:**

This paper presents an empirical study comparing visual-only versus geometry-grounded features in distilled radiance fields, alongside a proposed pose estimation framework (SPINE). While the finding that geometry-grounding can degrade semantic versatility is empirically interesting, the consensus among reviewers is that the work lacks the technical novelty and theoretical depth typically required for ICLR.

**Reviewer Concerns:**

The authors successfully addressed the concern regarding limited scope by adding evaluations for DUSt3R and MASt3R, satisfying the specific request of Reviewer x4tX. However, substantial concerns from Reviewers HjDG, mf1Y, and uFrD regarding the limited technical novelty of the SPINE framework (viewed as a combination of standard tools) and the lack of deeper mechanistic explanations for the observed "catastrophic forgetting" remain outstanding.

**Reviewer Scores:**

Reviewer x4tX likely would have raised their score to a 5 or 6 (Weak Accept) as they explicitly requested the mIoU metrics provided in the supplementary material. Reviewers HjDG, mf1Y, and uFrD would likely have maintained their lower scores (2-4), as their fundamental critiques regarding the paper being an "exploratory study" rather than a methodological advance were not fully resolved by the additional experiments.

---

### Decision · Program_Chairs · 2026-01-26

Reject